# Comparison of Tree-Structured Parzen Estimator Optimization in Three Typical Neural Network Models for Landslide Susceptibility Assessment

**Guangzhi Rong** [1,2,3], **Kaiwei Li** [1,2,3], **Yulin Su** [1,2,3], **Zhijun Tong** [1,2,3], **Xingpeng Liu** [1,2,3], **Jiquan Zhang** [1,2,3,*] , **Yichen Zhang** [4] **and Tiantao Li** [5,6]

1   School of Environment, Northeast Normal University, Changchun 130024, China; ronggz728@nenu.edu.cn (G.R.); likw395@nenu.edu.cn (K.L.); suyl686@nenu.edu.cn (Y.S.); gis@nenu.edu.cn (Z.T.); liuxp912@nenu.edu.cn (X.L.)
2   Key Laboratory for Vegetation Ecology, Ministry of Education, Changchun 130117, China
3   State Environmental Protection Key Laboratory of Wetland Ecology and Vegetation Restoration, Northeast Normal University, Changchun 130117, China
4   Changchun Institute of Technology, School of Emergency Management, Changchun 130012, China; weifenfangcheng@tom.com
5   College of Environment and Civil Engineering, Chengdu University of Technology, Chengdu 610059, China; litiantao18@cdut.edu.cn
6   State Key Laboratory of Geohazard Prevention and Geo Environment Protection, Chengdu University of Technology, Chengdu 610059, China
*   Correspondence: zhangjq022@nenu.edu.cn; Tel.: +86-1359-608-6467

**Abstract:** Landslides pose a constant threat to the lives and property of mountain people and may also cause geomorphological destruction such as soil and water loss, vegetation destruction, and land cover change. Landslide susceptibility assessment (LSA) is a key component of landslide risk evaluation. There are many related studies, but few analyses and comparisons of models for optimization. This paper aims to introduce the Tree-structured Parzen Estimator (TPE) algorithm for hyperparameter optimization of three typical neural network models for LSA in Shuicheng County, China, as an example, and to compare the differences of predictive ability among the models in order to achieve higher application performance. First, 17 influencing factors of landslide multiple data sources were selected for spatial prediction, hybrid ensemble oversampling and undersampling techniques were used to address the imbalanced sample and small sample size problem, and the samples were randomly divided into a training set and validation set. Second, deep neural network (DNN), recurrent neural network (RNN), and convolutional neural network (CNN) models were adopted to predict the regional landslides susceptibility, and the TPE algorithm was used to optimize the hyperparameters respectively to improve the assessment capacity. Finally, to compare the differences and optimization effects of these models, several objective measures were applied for validation. The results show that the high-susceptibility regions mostly distributed in bands along fault zones, where the lithology is mostly claystone, sandstone, and basalt. The DNN, RNN, and CNN models all perform well in LSA, especially the RNN model. The TPE optimization significantly improves the accuracy of the DNN and CNN (3.92% and 1.52%, respectively), but does not improve the performance of the RNN. In summary, our proposed RNN model and TPE-optimized DNN and CNN model have robust predictive capability for landslide susceptibility in the study area and can also be applied to other areas containing similar geological conditions.

**Keywords:** landslide susceptibility assessment; deep neural network; recurrent neural network; convolutional neural network; hyperparameter optimization; tree-structured Parzen estimator algorithm

## 1. Introduction

Landslides are the most common natural hazards in mountainous areas, and once occurred, landslides may cause the destruction of roads and houses, change of large

land use, and even bring death and huge economic losses, which seriously affect the sustainable development of society and economy [1,2]. Landslides are caused by the combined effects of internal and external dynamic geological action or human engineering activities, resulting in some degree of damage to the geological environment. China is a country where landslides occur very frequently and the damage is extremely serious. According to the 2017 China Land, Mineral and Marine Resources Statistical Bulletin released by the Ministry of Natural Resources of China, just in 2017, 7122 landslides caused 327 deaths, 25 persons missing, 173 injuries, and direct economic losses of up to CNY 3.537 billion in China.

There are about one million historical landslide sites, including topple, slide, debris flow, ground subsidence, and other types. Among them, topples, slides, and debris flows constitute 80% of the whole landslides [3]. Topple is the crumbling and rolling of a rock and soil body on a slope after it has been suddenly detached by gravity, slide is the overall downward sliding of a rock body on a slope under the action of gravity for some reason along a certain weak surface or zone of weakness, debris flow is a special type of flood with large quantities of sediment, rocks and other solid material conditions formed by precipitation. Although the trigger conditions and thresholds of the three landslide types are different, the geological and hydrological conditions (susceptibility factors) of the areas where they may occur are extremely similar, and they are more frequent and hazardous. Therefore, topples, slides, and debris flows are integrated to represent landslide for susceptibility assessment in this paper.

Landslide risk assessment is a comprehensive analysis and evaluation of potential losses from disasters based on landslides and integrated natural, social, and economic factors, and it results in regional disaster reduction planning and providing technical support with operability [4,5]. Landslide susceptibility assessment (LSA) is a key component of the landslide risk evaluation. There are many related studies that assess the susceptibility or risk of landslides on a national scale, such as China [6,7], Portugal [8], Iran [9], and New Zealand [10], and even on a global scale [11,12]. For national scale LSAs, these studies use conventional models such as logistic regression (LR), random forest (RF), etc., and even incorporate local policy orientations and considerations of the physical vulnerability of buildings. Additionally, at the global scale, there are studies of landslide non-susceptibility mapping, this literature offers a variety of ideas for large scale LSA studies. However, the analysis is more often carried out for smaller scales such as counties and cities [13–17]. These studies allow for more targeted development of new methods and models to assist local disaster management authorities.

In the past three decades, LSA methods and theories have made great progress, especially in shifting from qualitative analysis to quantitative assessment, which is attributable to the development of spatial and information technology, making the originally complex arithmetic process and tedious data acquisition easy to operate. With the continuous development of mathematical models and computer technology, the research methods of regional LSA are still being innovated. The majority of conventional studies are mathematically and statistically based methods [18–21]. Some researchers used mathematical statistical models such as hierarchical analysis, interval rough number-hierarchical analysis, entropy power method-hierarchical analysis to evaluate and analyze the distribution and development characteristics of landslides [22], others used information value, and weight of evidence methods to determine the landslide susceptibility, and used validation methods such as applying a receiver operating characteristic curve (ROC), proportional correct classification, and seed cell area index (SCAI) for evaluation [23,24]. Frequency ratio method was also used the to assess the response of individual elements to the landslide incidence [25], even others have used the series-parallel model of physics to construct a weighted indicator system [26].



With the continuous iteration and development of computer science and technology, many studies have started to introduce machine learning algorithms in the study of LSA [27,28]. The LR is one of the most classic machine learning models that has been introduced by many researchers into the study of landslide risk evaluation [29–31]. The decision tree (DT) and RF were often compared for the effectiveness of models in LSA [32]. Neural network models, especially the basic artificial neural network (ANN) model was used to map landslide susceptibility and obtained very high accuracy results [33,34]. In addition, many other machine learning methods are already available for risk evaluation studies of landslides, such as naive Bayesian [35], gradient boosting decision tree model [36], support vector machine model [37,38], genetic algorithm [39], recurrent neural network (RNN) model [40], and convolutional neural network (CNN) model [41], etc. Moreover, there are many comparisons of different methods and some hybrid algorithms in LSA [42–44]. However, one of the most important steps of machine learning models is the tuning and optimization of parameters, which is the focus and difficulty of future research [45]. In this paper, we select the topic to focus on this literature gap and choose the latest three typical neural network models (deep neural network (DNN), RNN, CNN) for hyperparametric optimization research in order to fill certain academic gaps and broaden the horizon and direction of LSA research.

For this work, the historical landslide data, geographical data, topographic data, and telemetric data were gathered to build an integrated multi-source database. DNN, RNN, and CNN models were used to assess the susceptibility of landslides for Shuicheng County, China, and the performance of these three typical neural network models was compared by various methods and indices. Based on the three models, the Three-structured Parzen Estimator (TPE) algorithm in Bayesian optimization was introduced to adjust the initial parameters (hyperparameters) of these models for achieving optimized performance of three typical neural network models in LSA. For the validation analysis of the models, we used the Accuracy, Precision, Recall, F-value, Matthews correlation coefficient (MCC), Kappa value, ROC curve, and SCAI to evaluate the performance of different neural network models and their TPE optimization in LSA from multiple perspectives. Based on the above methods, we mapped the LSMs for reference in preventing and mitigating landslides. This paper innovatively discusses the application of three typical neural network models in LSA and introduces the TPE algorithm to optimize the hyperparameters of the models in order to achieve better accuracy. The new techniques we propose are scientific and feasible in LSA and have robust prediction ability and application prospects. This study can provide guidance and suggestions for local disaster management decision-making, contribute to the continuous innovation and development of LSA research, and has certain theoretical and practical significance.

## 2. Study Area and Data

### 2.1. Study Area

Shuicheng County belongs to the central region of the Yunnan-Guizhou Plateau in China, with an approximate area totaling 3605 km$^2$ and a resident population of about 754,900. Its elevation range is from 633 to 2863 m, and about 32.5% of areas have a slope above 20° (Figure 1). For landslides, extreme precipitation is the main triggering and inducing factor. Sudden extreme precipitation in mountainous areas induces regional or basin-based cluster landslides such as topples, slides, and debris flows which can cause serious impact on social production life and ecological environment. Shuicheng County belongs to subtropical monsoon climate, with abundant and frequent precipitation, often accompanied by heavy rainfall. Furthermore, it also belongs to karst landscape, where surface water easily seeps and the moisture content of the soil is high. Shuicheng County is a concentrated and high-incidence area of landslides and is one of the landslide-prone and serious counties in Guizhou Province. Hence, it is particularly important to carry out risk assessment of extreme precipitation-induced landslides in Shuicheng County, and the susceptibility assessment is the most essential content of it.

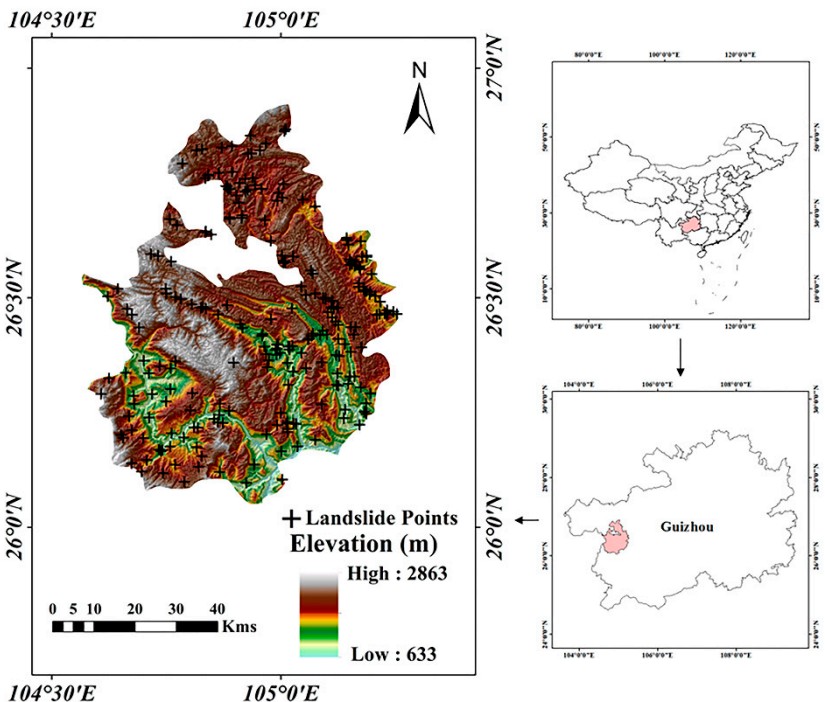

**Figure 1.** Location and landslide points of Shuicheng County.

On 23 July 2019, a mega-landslide that killed 52 people occurred in the Jichang Town, Shuicheng County [46,47]. Figure 2 shows the Google Earth satellite maps prior to and following the landslide, and that can be seen more than 1 year after the landslide there is still a very significant impact on natural factors such as vegetation and geomorphology, as well as roads, houses, and other building sites.

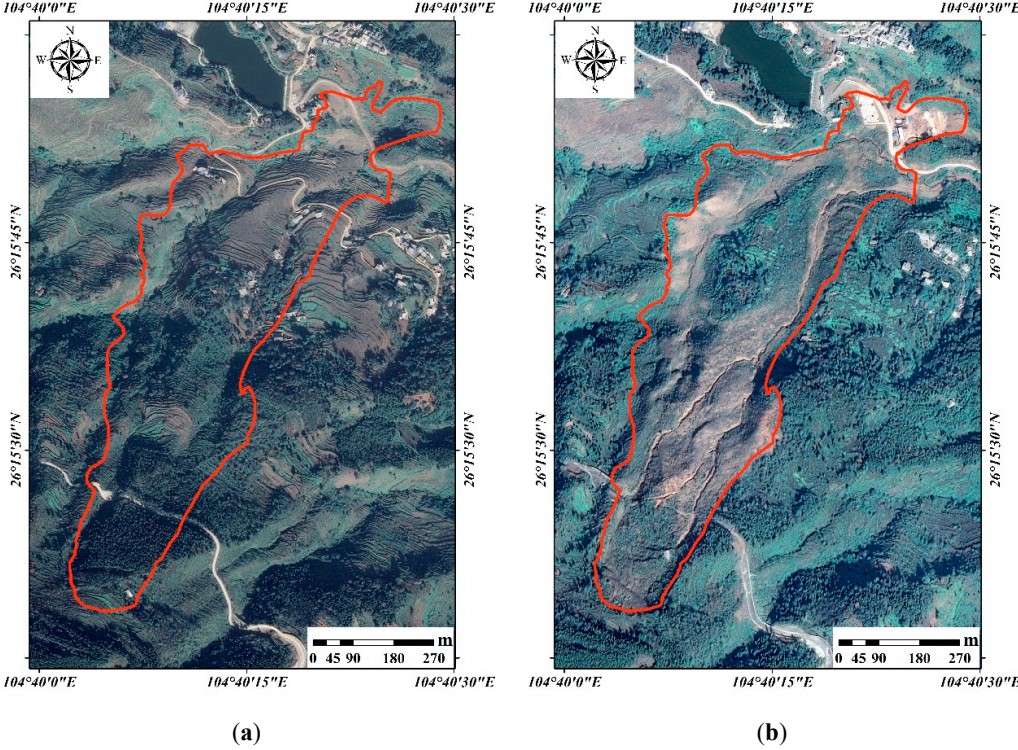

(**a**)                                                  (**b**)

**Figure 2.** Pre-sliding and post-sliding Google Earth satellite maps of the Jichang landslide: (**a**) before the slide (14 November 2018) and (**b**) aerial image after the slide (11 November 2020).

## 2.2. Landslide Historical Inventory

In this paper, we collected the landslide historical inventories recorded by China Geological Survey [48]. Additionally, the three most frequent and serious forms of landslides, topples, slides, and debris flows, were extracted, and integrated with remote sensing images and field survey, the centroids of landslide scarp of 240 historical landslide points were finally identified and stored in the database, which has been proved the best landslide sampling strategy [49].

## 2.3. Landslides Susceptibility Influencing Factors

The susceptibility is characterized as the spatial probability of occurrence of landslides, and the selection of the influencing factors is significantly important to LSA. Combining relevant studies and the accessibility of factors, we finally identified 17 factors. We input these influencing factors into a uniform format database, according to the Digital Elevation Model (DEM) map pixel size, all the factor's pixel sizes were set to $30 \times 30$ m, regardless of the initial data format.

Lithology is fundamental to the development of landslides and affects the shear strength and water leakage of slopes [50]. The lithology in Shuicheng County can be classified into five categories, Basalt, Claystone, Dolomite, Limestone, and Sandstone. Geological age can reflect the degree of lithological development. Meanwhile, the distance to the faults can reflect the active degree of geological structure. This part of vector data was digitized from geological maps provided by the China Geology Survey. Topographic factors are another major predisposing factor for landslides [2]. Elevation is a measure of the absolute degree of elevation of regional terrain; slope is an important factor for landslide development, especially for rockfalls and landslides. Slope and geotechnical stability are not simply linearly related, it always works in conjunction with slope height, geotechnical combination, slope structure, and other factors. Aspect generally combines with other factors to form the slope structure, which in turn affects the development of landslides. For landslides induced by extreme rainfall, the aspect affects the insolation and rainfall. The slope type mainly includes concave, convex, and linear, etc. The degree of rainfall infiltration into the slope body varies with the slope type, which can be characterized by plan curvature and profile curvature. Land cover affects soil erosion, especially the degree of vegetation cover, which can be expressed by the normalized difference vegetation index (NDVI) [51,52]. The land cover was downloaded at Finer Resolution Observation and Monitoring of Global Land Cover (FROM-GLC) and the NDVI was calculated from the near infrared and red band values in Landsat8 OLI satellite remote sensing digital images shot in April 2018:

$$NDVI = \frac{Near\ infrared - Red}{Near\ infrared + Red} \tag{1}$$

Roads can reflect the influence of human activities on geological formations, so the distance from the road was also selected as an influencing factor. Terrestrial hydrology mainly refers to the development and distribution of river valleys, etc. Research shows that the development of landslides is strongly linked to the river system configuration, because river system integrally reflects the development of the free surface, the density of gully, and some characteristics of slope. The river also has an erosion effect, mainly downward cutting erosion, lateral hollowing erosion, and wave action, erosion will carry away stones and clods at the slope toe, forming the free plane, and providing favorable topographic conditions for landslides. Meanwhile, the rise and fall of river level will affect the fluctuation of groundwater level, which will affect the stability of the slope. Based on the above analysis, distance to rivers, average annual precipitation (AAP), and four hydrological indices were selected for landslide studies [31]. Among them, the distance to rivers was obtained by creating multi-loop buffer, the AAP was obtained by collecting the AAP of seven meteorological stations around the study area from 1981 to 2018, and then interpolating by the inverse distance weights, and the four hydrological indices were calculated as shown below, and the four hydrological indices consisting of Stream

Power Index (SPI), Sediment Transport Index (STI), Topographic Relief Index (TRI), and Topographic Wetness Index (TWI) [53,54], these indices were calculated as shown below:

$$SPI = A_S \times \tan \beta \tag{2}$$

$$STI = \left( \frac{A_S}{22.13} \right)^{0.6} \times \left( \frac{\sin \beta}{0.0896} \right)^{1.3} \tag{3}$$

$$TRI = DEM_{MAX} - DEM_{MIN} \tag{4}$$

$$TWI = \ln \frac{A_S}{\tan \beta} \tag{5}$$

where $A_S$ represents the catchment area (m$^2$/m), $\beta$ is the slope [55], $DEM_{MAX}$ and $DEM_{MIN}$ are the max and min DEM value surrounding every pixel, respectively.

The classification of each influencing factor value is shown in Table 1.

**Table 1.** The classification of each influencing factor value.

| Influencing Factors | Variable Type | Resolution | Unit | Class | | | | |
|---|---|---|---|---|---|---|---|---|
| | | | | 5 | 4 | 3 | 2 | 1 |
| Lithology | Discrete | - | - | Claystone | Basalt | Sandstone | Dolomite | Limestone |
| Geological age | Discrete | - | - | Devonian | Carboniferous | Permian | Triassic | Jurassic |
| Distance to Faults | Continuous | - | m | <500 | 500~700 | 700~1000 | 1000~2000 | >2000 |
| Elevation | Continuous | 30 × 30 m | m | 2079~2863 | 1827~2079 | 1570~1827 | 1248~1570 | 633~1248 |
| Slope | Continuous | 30 × 30 m | ° | 40.05~74.74 | 28.58~40.05 | 19.70~28.58 | 11.56~19.70 | 0~11.56 |
| Aspect | Discrete | 30 × 30 m | - | Sunny slope | Semi-sunny slope | Flat | Semi-shady slope | Shady slope |
| Plan curvature | Discrete | 30 × 30 m | - | >0.33 | 0.10~0.33 | −0.10~0.10 | −0.33~−0.10 | <−0.33 |
| Profile curvature | Discrete | 30 × 30 m | - | >0.42 | 0.11~0.42 | −0.11~0.11 | −0.42~−0.11 | <−0.42 |
| Land cover | Discrete | 30 × 30 m | - | Impervious surface | Cropland, bare land | Shrubland | Forest, grassland | Wetland, water, snow/ice |
| NDVI | Continuous | 30 × 30 m | - | <0.111 | 0.111~0.162 | 0.162~0.201 | 0.201~0.247 | >0.247 |
| Distance to roads | Continuous | - | m | <500 | 500~700 | 700~1000 | 1000~2000 | >2000 |
| Distance to rivers | Continuous | - | m | <500 | 500~700 | 700~1000 | 1000~2000 | >2000 |
| AAP | Continuous | 30 × 30 m | mm | 1272~1353 | 1212~1272 | 1150~1212 | 1080~1150 | 981~1080 |
| SPI | Continuous | 30 × 30 m | - | >1000 | 500~1000 | 100~500 | 0~100 | 0 |
| STI | Continuous | 30 × 30 m | - | >20 | 10~20 | 1~10 | 0~1 | 0 |
| TRI | Continuous | 30 × 30 m | - | 91~342 | 58~91 | 38~58 | 22~38 | 0~22 |
| TWI | Continuous | 30 × 30 m | - | 11.87~23.50 | 8.73~11.87 | 6.61~8.73 | 5.00~6.61 | 1.85~5.00 |

All the maps of the spatial distribution of influencing factors of landslides are shown in Figure 3.

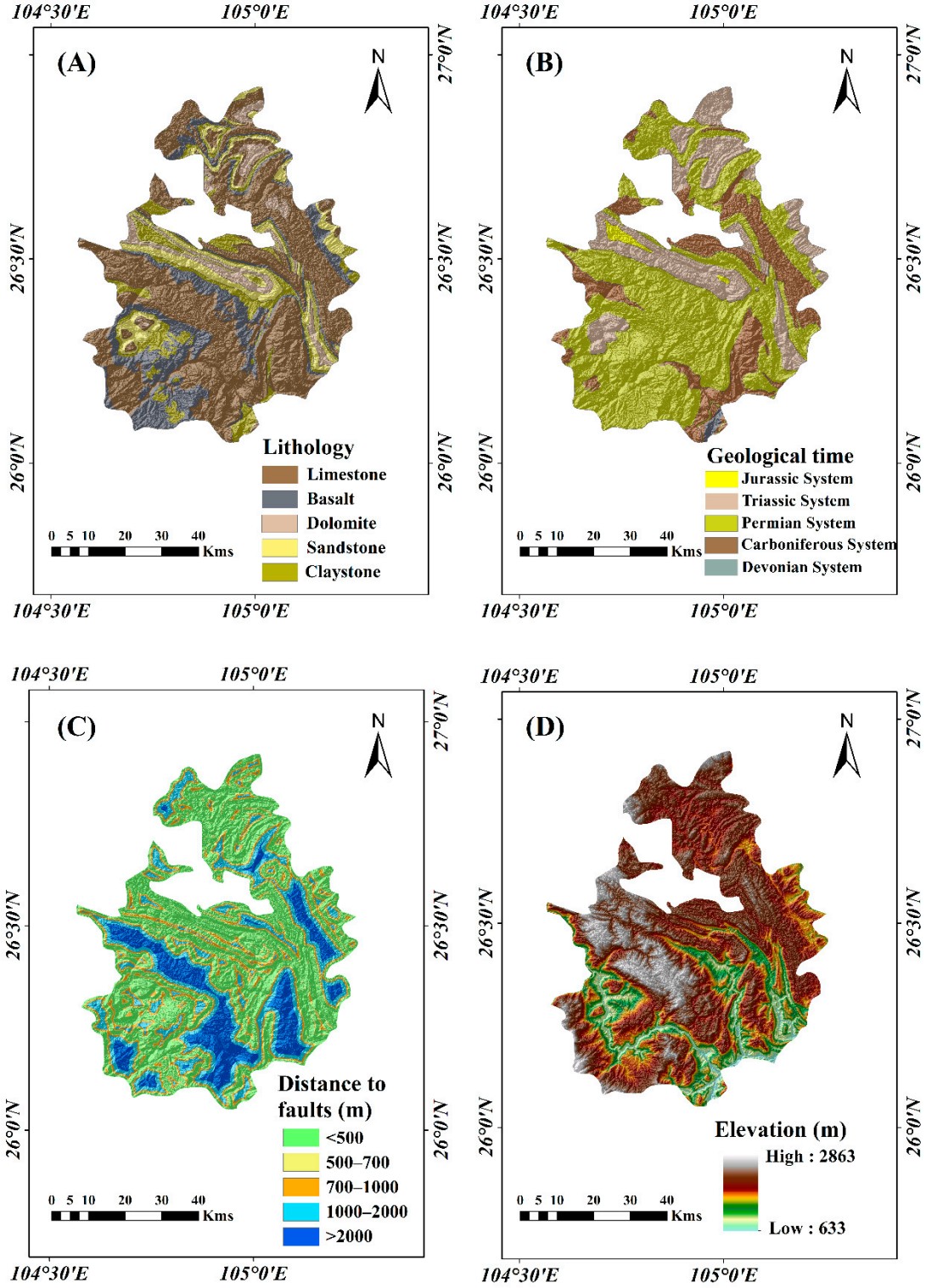

**Figure 3.** *Cont.*

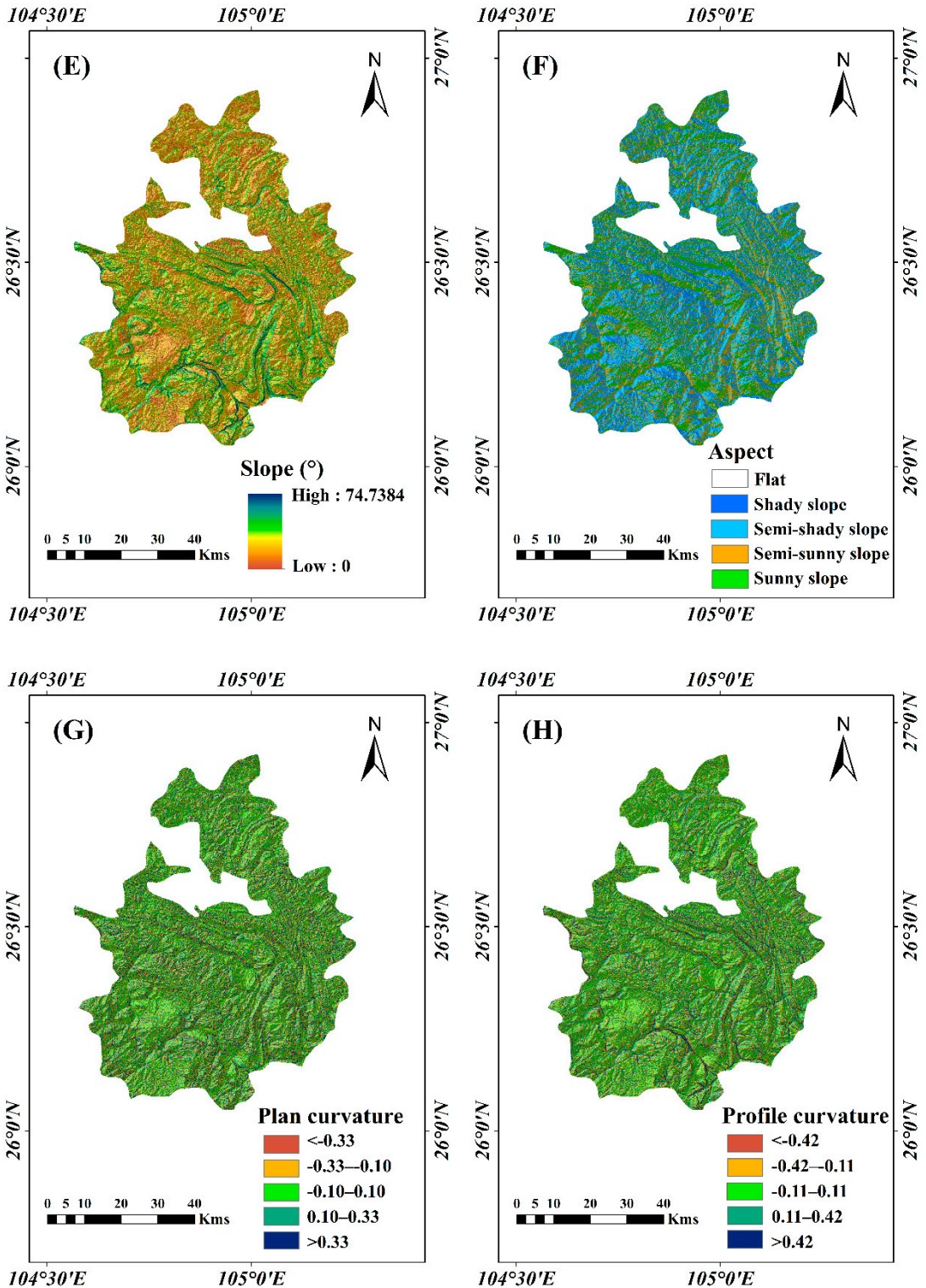

**Figure 3.** *Cont.*

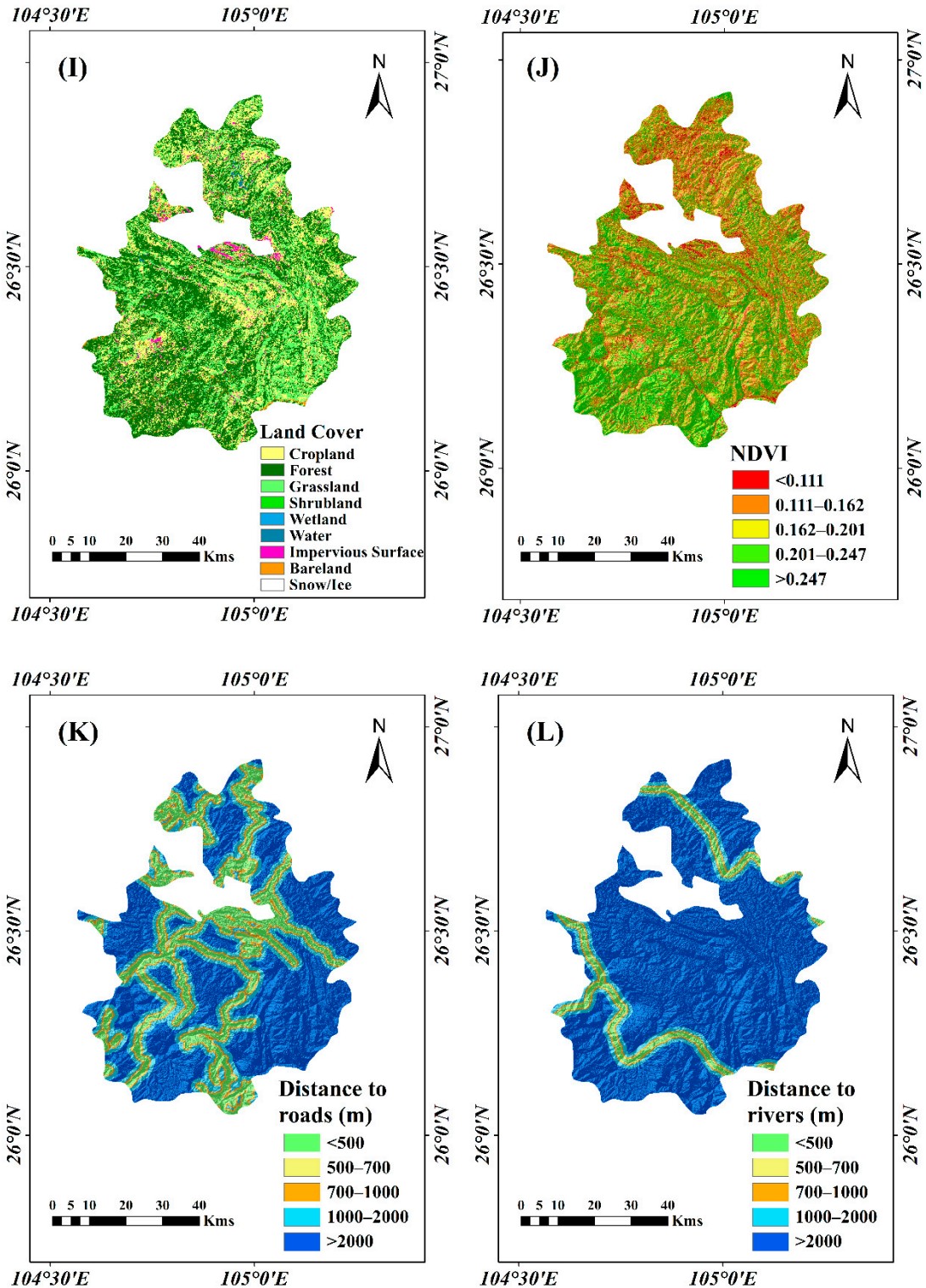

**Figure 3.** *Cont.*

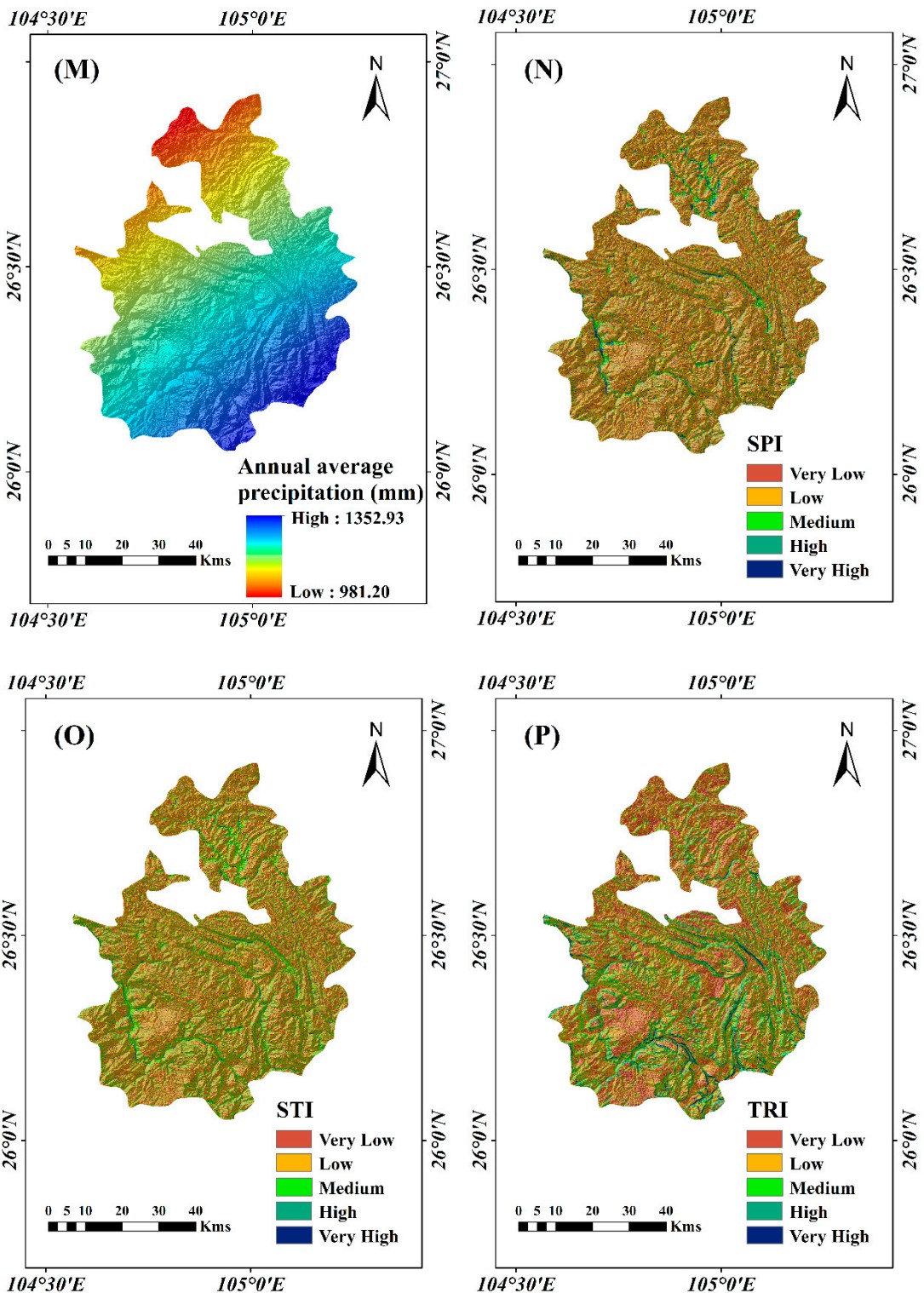

**Figure 3.** *Cont.*

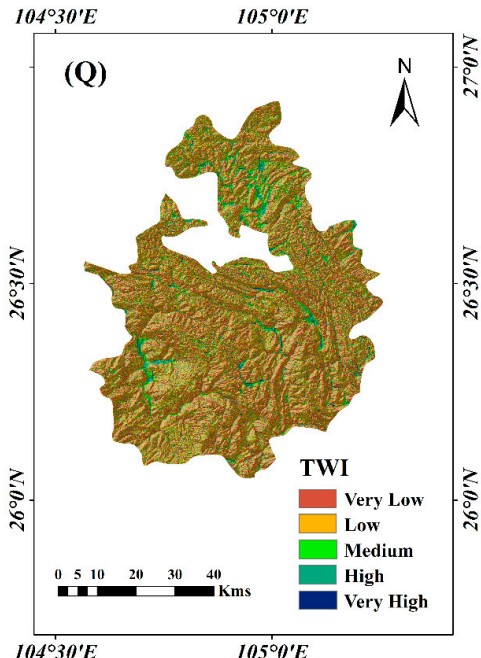

**Figure 3.** Maps of the spatial distribution of influencing factors. (**A**) Lithology, (**B**) geological age, (**C**) distance to faults, (**D**) elevation, (**E**) slope, (**F**) aspect, (**G**) plan curvature, (**H**) profile curvature, (**I**) land cover, (**J**) NDVI, (**K**) distance to roads, (**L**) distance to rivers, (**M**) annual average precipitation, (**N**) SPI (**O**) STI, (**P**) TRI, (**Q**) TWI.

## 3. Methods

Our research of LSA can be separated in the five stages as follows, which can be observed in Figure 4. Stage A is data collection; we collected data of landslide historical inventory and the landslide susceptibility influencing factors through multiple sources. Stage B is data processing; we divided the study area into grids, unified data to the same pixel size, and prepared the samples. Stage C is model construction; we used the DNN, RNN, and CNN models to train and generate LSMs. Stage D is TPE optimization, we used the TPE optimized DNN (DNN_TPE), RNN (RNN_TPE), and CNN (CNN_TPE) to train the models and generate LSMs as well. Stage E is model validation and comparison; we validated and compared the performance and the TPE optimization effect of different neural network models by multiple methods.

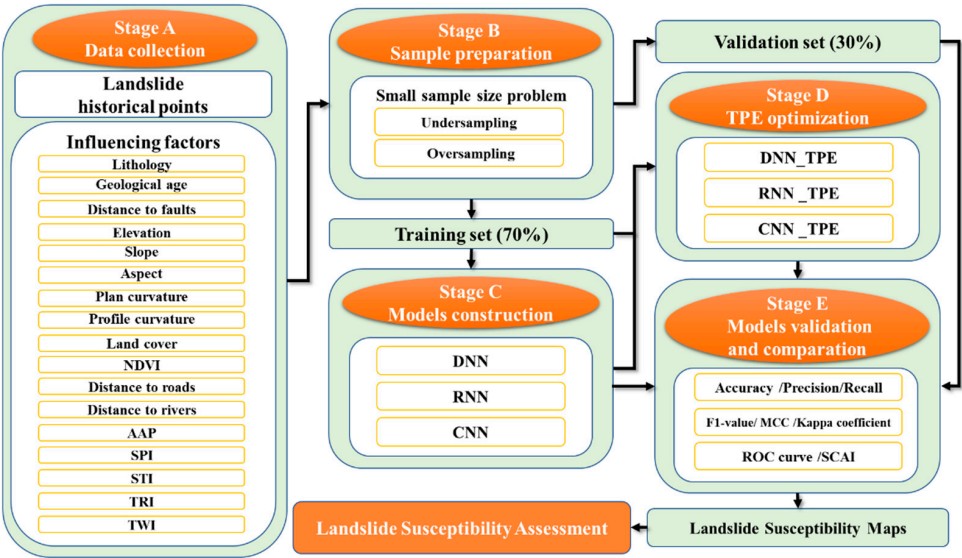

**Figure 4.** Flowchart of the proposed landslide susceptibility assessment framework.

*3.1. Data Pretreatment*

3.1.1. Geodatabase Construction

First, the factors were classified into 5 categories, where continuous variants used the Natural Breaks Method (NBM), and discrete variants were ranked by calculating the ratio of historical landslide points (*R*) to the area for each category:

$$R = \frac{X_{ij}/X_A}{S_{ij}/S_A} \times 100\% \tag{6}$$

where $S_{ij}$ and $S_A$ represent the area of category *j* of factor *i* and the study area, respectively. $X_{ij}$ and $X_A$ are the number of historical landslide points in $S_{ij}$ and $S_A$, respectively. *R* actually represents the amount of information in each category, and the higher the *R* value, the higher the category rank.

3.1.2. Sample Selection

In the neural network modeling process, the number of positive samples (landslide points) and negative samples (non-landslide points) should not be unbalanced by orders of magnitude, because when the data are extremely unbalanced, samples from the majority category are easier to predict, and the prediction performance for minority category is poorer. Meanwhile, a total of 240 historical landslide points in Shuicheng County were identified in this study. Too few numbers may lead to poor model prediction and cannot correctly reflect the vulnerability of landslides in the study area, while too many may cause overfitting of the model. After several tests, when the landslide points are doubled and then an equivalent number of non-landslide points are selected as samples, a certain accuracy can be maintained without overfitting.

Considering the above, this paper used the hybrid ensemble oversampling and under-sampling techniques for sample selection. The specific steps are as follows: (1) 240 non-landslide points were selected using random undersampling and repeated twice, 480 negative samples were obtained; (2) these 480 non-landslide points and the 240 landslide points were selected as input data and the Borderline-Synthetic Minority Over-sampling Technique (Borderline-SMOTE) algorithm was used to oversample the positive samples which is an enhanced method of SMOTE [56]. The principle of SMOTE is by selecting a minority class sample A, choosing a sample B from its nearest neighbors, and then generating a new minority class sample randomly on the line of the two points, while Borderline-SMOTE is based on this, but only safe samples (A, B are the same class) are selected for the sample synthesis. In this paper, a new 240 landslide points were generated; (3) 70% of the positive and negative samples were chosen at random for training while the remaining 30% were used for validation, respectively.

*3.2. Multi-Collinearity Analysis*

Multi-collinearity analysis is a prerequisite to test whether multi-dimensional factors can be used simultaneously. In this paper, variance inflation factor (*VIF*) was selected as the determination conditions:

$$VIF = \frac{1}{1 - R_j^2} \tag{7}$$

where $R_j^2$ is determinable co-efficient of the auxiliary regression model of the explaining factors $X_j$ on the others. The closer the *VIF* is to 1, the weaker the multi-collinearity. Experience shows that $VIF \geq 10$ indicates severe multi-collinearity between the variables and the remaining variables, and this multi-collinearity may overly affect the least squares estimates. Tolerance is the inverse of *VIF*, which means that severe multi-collinearity exists when Tolerance $\leq 0.1$.

### 3.3. DNN Model

DNN can be understood as neural networks with many hidden layers [57]. DNN extends the simple perceptron by: (1) adding multi-layer hidden layers to enhance the expressiveness of the model; (2) the output layer neurons can be more than one and can have multiple outputs, so that the model can be flexibly applied to classification, regression, dimensionality reduction and clustering, etc.; (3) the activation function can be extended. The activation function of the perceptron is sign(z), which is simple but has limited processing power, while the neural network generally uses Sigmoid, tanh, ReLU, softplus, softmax, etc., to add nonlinear factors, which can improve the expressiveness of the model. The structure of the DNN constructed in this paper is shown as Figure 5.

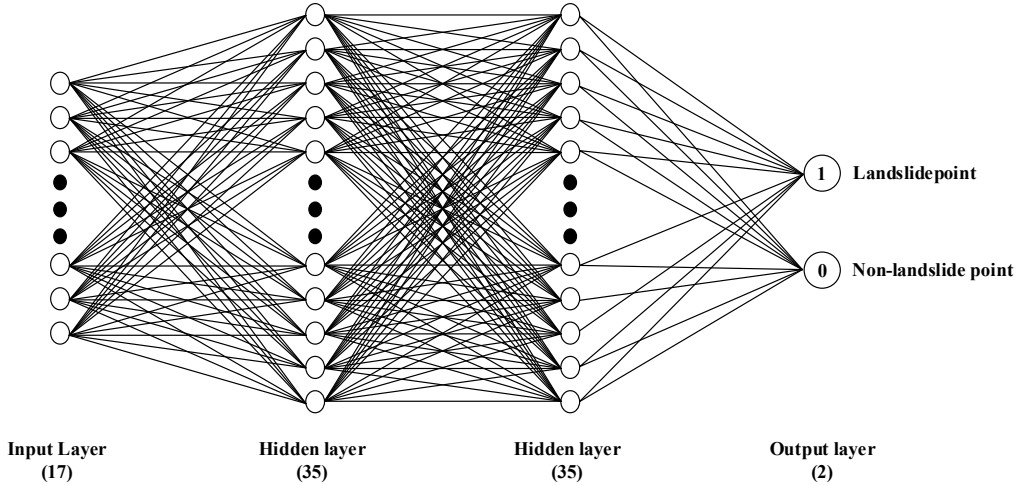

**Figure 5.** Structure of the constructed deep neural network (DNN) model.

### 3.4. RNN Model

RNN is a special neural network structure that not only considers the input at the previous moment, but also creates a "memory" of the previous content. In other words, the present output is correlated with the preceding output as well [58]. The specific expression is that the network remembers the prior information and applies them in the computation of the present export, which means that the nodes between the hidden layers are no longer connectionless but connected, and the input of the hidden layers also contains the output of the hidden layers in the preceding moment. Based on this property, RNNs are commonly used in speech recognition research [59]. Figure 6 shows the layer unfolding of the hidden layers of the RNN model.

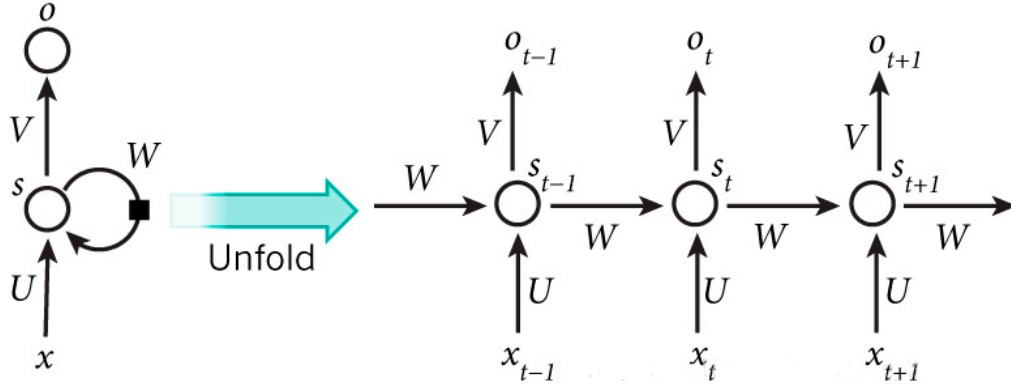

**Figure 6.** The layer unfolding of the hidden layers of the recurrent neural network (RNN) model.

$t-1, t, t+1$ denote the time series. $x$ denotes the input sample, $h_t$ denotes the hidden state vector at time $t$, $S_t$ denotes the memory of the sample at time $t$, $S_t = f(W \times S_{t-1} + U \times x_t)$. $W$ denotes the weight of the input sample, $U$ denotes the weight of the input sample at this moment, and $V$ denotes the weight of the output sample. When $t = 1$, the general initialization input $S_0 = 0$, random initialization $W, U, V$, and proceed to the following Equation:

$$h_1 = Ux_1 + Ws_0$$
$$s_1 = f(h_1) \tag{8}$$
$$o_1 = g(Vs_1)$$

where, $f(x)$ and $g(x)$ are both activation functions, $f(x)$ can be Tanh, Relu, Sigmoid and other activation functions, $g(x)$ is usually used by Softmax. and so on, the final output value can be obtained as:

$$h_t = Ux_1 + Ws_{t-1}$$
$$s_t = f(h_t) \tag{9}$$
$$o_t = g(Vs_t)$$

There are various variants of RNN models that may have surpassed performance of the basic RNN model [40,60,61]. However, since this paper aims to compare the applications of different typical neural network models, the basic RNN model was selected and the default neurons in the hidden layer were set to 50, and the structure of the RNN model is shown in Figure 7.

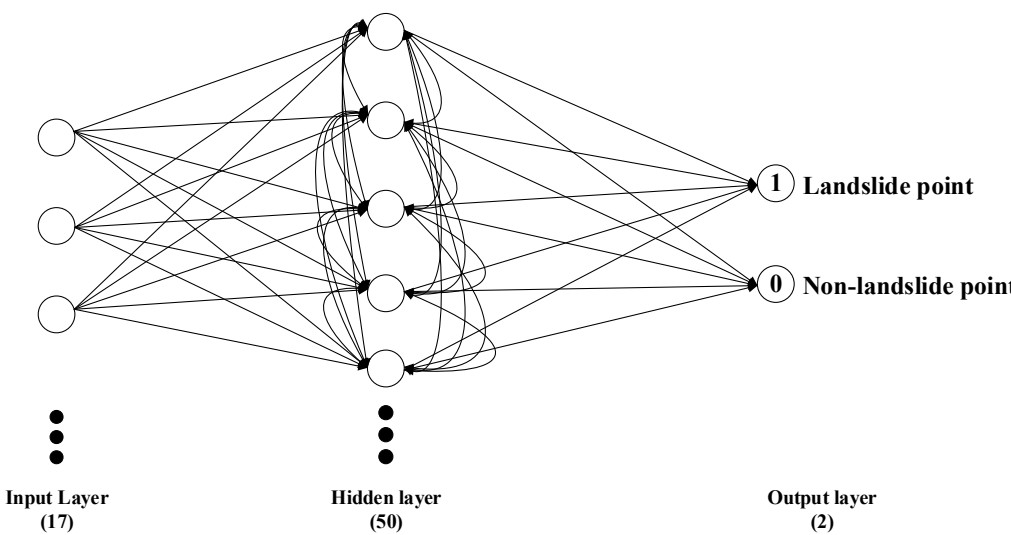

**Figure 7.** Structure of the constructed RNN model.

*3.5. CNN Model*

CNN is essentially a multilayer perceptron proposed by Yann Lecun of New York University in 1998 [62]. CNN is characterized by local connectivity and shared weights, which decreases weight counts making this network easy to optimize, while reducing the model sophistication, that is, risks for overfitting. The special feature of CNN construction is that it has a unique convolutional layer and pooling layer. Convolutional layer is functioned to extract features, in the convolution operation, a matrix of size F × F (F × 1 in one dimension) is set, called the filter or convolution kernel, and the matrix size is receptive field. The interior of the convolutional layer contains multiple convolutional kernels, and each element that makes up the convolutional kernel is associated with a weight and a bias. Every neuron within a convolutional layer is connected to several others near its position in the preceding layer, and the region magnitude is determined by the receptive field. The convolution

kernel works by regularly sweeping through the input features, multiplying and summing matrix elements within the convolution kernel, and superimposing the bias. Then, the exported feature graph is delivered into the pooling layer to be used for feature selection and informational filtration. The pooling layer includes pre-defined pooling functions, Max pooling and average sampling are the most common. It can replace the value for a point within a feature map with the statistical value of the adjacent areas. The pooling layer selects the pooling region at the identical manner as the convolutional kernel scans the feature map, determined by the padding, size of pooling and step. The pooling layer is equivalent to converting a higher resolution image into a lower resolution image, it also reduces the node count in the final fully connected layer, thus reducing parameters of the entire neural network and hence decreases overfitting risk. The fully connected layer is the final component of the CNN hidden layer, which equals the hidden layer in a classical feed-forward ANN model. The fully connected layer is responsible for transmitting information to the output layer where the feature maps will lose their spatial topologies, be extended as vectors, and passed through the activation function. As the most excellent and popular neural network model in latest decade, CNN was extensively adopted for various fields, especially image recognition [63,64].

For LSA, based on the influence factor rank of each pixel, each sample can be made into a $17 \times 1$ array format as the input layer, therefore, in this paper, a one-dimensional CNN, which is often applied to the data processing of sequence class, was used to construct the model. The specific CNN structure is shown in Figure 8. This structure is referred to the one-dimensional CNN presented by Wang et al. [41]. The input layer is the dimension $n$ of the factor, which is $17 \times 1$, the initial value of the convolution kernel $m$ is set to 3, and $N$ feature vectors of length $(n - m + 1)$ are obtained, with $N$ set to 20. The size of the maximum pooling layer is $a \times 1$, and the initial value of $a$ is set to 2. The result consists of $N$ vectors of length $(n - m + 1)/a$. Then, a fully connected layer with 50 neuron units is set up for the extracted features. Finally, we set 2 neurons in the output layer to achieve the problem of binary classification.

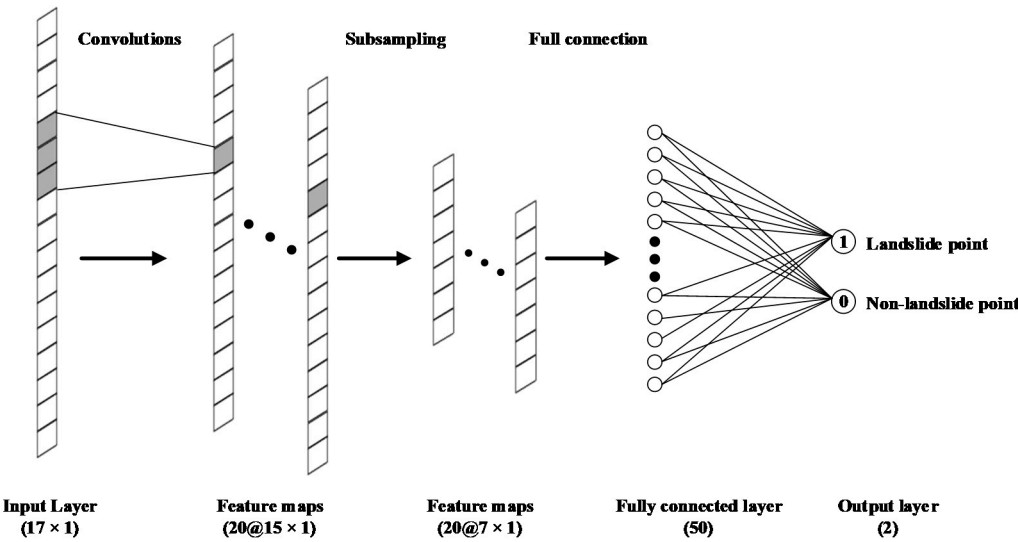

**Figure 8.** Structure of the constructed convolutional neural network (CNN) model.

### 3.6. TPE Optimization

Hyperparameter optimization has been extremely important for machine learning models, especially for neural network models which are typically black box models [65]. Since it cannot intervene during model training, tuning hyperparameters before the model runs formally becomes an important means to enable the improvement of model precision. From the initial manual tuning to the later evolution of grid and random search, it was very time consuming and inefficient [54]. Based on the idea of accuracy and efficiency, many



methods for automatic tuning of parameters were later generated. Bayesian optimization is a function minimization method using a model to find the value that minimizes the objective function [66]. It is highly performant and very time-efficient since it refers to the previous evaluation results when trying the next set of hyperparameters.

TPE is a Bayesian optimization algorithm proposed by [67], to learn hyperparameter models using the Gaussian Mixture Model. Firstly, the concept of conditional probability from Bayes theory is introduced. $p(x|y)$ is the conditional probability that the hyperparameter is $x$ when the model loss is $y$. In the first step, we select a threshold $y^*$ for the loss based on the available data, e.g., according to the median. Two probability densities $\ell(x)$ and $g(x)$ are learned for data greater than the threshold and less than the threshold, respectively.

$$p(x|y) = \begin{cases} \ell(x) & if \ y < y^* \\ g(x) & if \ y \geq y^* \end{cases} \tag{10}$$

where $\ell(x)$ is the density formed by using the observations $\left\{x^{(i)}\right\}$ such that corresponding loss $f\left(x^{(i)}\right)$ was less than $y^*$ and $g(x)$ is the density formed by using the remaining observations.

The parametrization of $p(x,y)$ as $p(y)p(x|y)$ in the TPE algorithm was chosen to facilitate the optimization of Expected Improvement (*EI*).

$$EI_{y^*}(x) = \int_{-\infty}^{y^*} (y^* - y)p(y \mid x)dy = \int_{-\infty}^{y^*} (y^* - y)\frac{p(x \mid y)p(y)}{p(x)}dy \tag{11}$$

By construction, $\gamma = p(y < y^*)$, and $p(x) = \int_{\mathbb{R}} p(x \mid y)p(y)dy = \gamma\ell(x) + (1 - \gamma)g(x)$,

$$\int_{-\infty}^{y^*} (y^* - y)p(x \mid y)p(y)\mathrm{d}y = \ell(x)\int_{-\infty}^{y^*} (y^* - y)p(y)dy = \gamma y^*\ell(x) - \ell(x)\int_{-\infty}^{y^*} p(y)dy \tag{12}$$

The final $EI_{y^*}(x)$ can be expressed as:

$$EI_{y^*}(x) = \frac{\gamma y^*\ell(x) - \ell(x)\int_{-\infty}^{y^*} p(y)dy}{\gamma\ell(x) + (1 - \gamma)g(x)} \propto \left(\gamma + \frac{g(x)}{\ell(x)}(1 - \gamma)\right)^{-1} \tag{13}$$

Therefore, we can minimize $g(x)/\ell(x)$ to get a new $x^*$, then put $x^*$ back into the function and iterate again to fit $\ell(x)$ and $g(x)$, keep minimizing $g(x)/\ell(x)$ until we reach the predetermined number of iterations, and finally complete the optimization of the hyperparameters.

In this paper, we used the Hyperopt library in python 3.7 environment to complete the TPE optimization. There are four main components of TPE optimization: (1) objective function: we select the loss of the model using the set of hyperparameters on the validation set; (2) domain space: that is, the search range of the hyperparameters; (3) optimization algorithm: that is, the TPE algorithm; and (4) the result record. In these four items, the objective function and optimization algorithm have been determined. Additionally, the domain space is another important part of TPE optimization. The basic principle of choosing the search space is scientific and efficient, the space range should not be too large or too small. Too small optimization effect is not obvious, too large is easy to cause overfitting and long computing time and other problems. The domain space selected in this paper is based on the above principles and related research.

For the selection of parameters, although there are many parameters of neural network models, many of them are nested in their selected activation functions, and the default values of these parameters can be chosen, which have little impact on model optimization and are generally not considered in Bayesian optimization. Only hyperparameters that directly affect the structure and operation of the network, such as the number of neurons, dropout rate, convolutional kernel size, and class of activation function, are selected for adjustment.

The hyperparameters of DNN and RNN are the same, but there is a difference in the "units", which refers to the neuron count of the two fully connected layers in DNN, and the neuron count of the hidden layer in the RNN model. Dropout rate is an important way to reduce overfitting in neural network models, especially when the training samples are small [68]. "batch size" is the number of samples selected for training at one time, and is a means of batch processing of neural networks, which can greatly improve the learning efficiency by processing samples in batches, especially for large-scale samples. If "batch size" increases, the gradient becomes accurate, and after a certain degree, it is useless to increase the "Batch Size". "Epoch" refers to the number of times all samples in a neural network model are trained, the more epochs, the more adequate, but too many epochs also tend to cause overfitting. Therefore, we set the initial values of "batch size" and "epoch" of DNN and RNN to 50, while the domain space is 10–100. The essence of machine learning training is in minimizing the loss, and after we define the loss function, we need the optimizer to perform the gradient optimization, and the goal of optimization is the loss value $\theta$ in the network model. In this paper, the most common optimizer algorithm Adaptive Moment Estimate (Adam) was chosen as the initial value, and introduced Adamax, Stochastic Gradient Descent (SGD), Root Mean Square Prop (RMSProp), Adaptive gradient algorithm (Adagred), Adadelta, Nesterov-accelerated Adaptive Moment Estimation (Nadam) as the domain space of the optimization algorithm in the process of TPE optimization. Each optimization algorithm has its own advantages and disadvantages, so TPE optimization is needed to search and find the optimal algorithm.

The hyperparameters of CNN model are more complicated than DNN and RNN. Firstly, the number of convolutional kernels "Filter" is also the number of convolutional layer feature map, the initial value is set to 20, and the domain space is 10–100. The "Kernels size" is the size of the convolution kernel, and we set the search range from 1 to 9, and the domain space of "pooling size" from 2 to 5. In addition, "Units", "Dropout rate", "Epoch", and "Optimizers" take the same range of values as DNN.

The initial values of hyperparameters and their domain spaces for each model to perform TPE optimization are shown in Table 2.

**Table 2.** Hyperparameters of TPE optimization and their initial values and domain space of DNN, CNN, and RNN models.

| Model | Hyperparameter | Initial Value | Domain Space |
|---|---|---|---|
| DNN | Units | 35 | [10, 100] |
| | Dropout rate | 0 | [0, 0.50] |
| | Batch size | 1 | [1, 100] |
| | Epoch | 50 | [10, 100] |
| | Optimizers | Adam | [Adam, Adamax, Sgd, Rmsprop, Adagrad, Adadelta, Nadam] |
| RNN | Units | 50 | [10, 100] |
| | Dropout rate | 0 | [0, 0.50] |
| | Batch size | 1 | [1, 100] |
| | Epoch | 50 | [10, 100] |
| | Optimizers | Adam | [Adam, Adamax, Sgd, Rmsprop, Adagrad, Adadelta, Nadam] |
| CNN | Fitters | 20 | [10, 100] |
| | Kernel size | 3 | [1, 10] |
| | Pooling size | 2 | [1, 10] |
| | Units | 50 | [10, 100] |
| | Dropout rate | 0 | [0, 0.50] |
| | Epoch | 100 | [10, 200] |
| | Optimizers | Adam | [Adam, Adamax, Sgd, Rmsprop, Adagrad, Adadelta, Nadam] |

### 3.7. Model Validation Methods

The goodness of the model needs to be judged by evaluating. In this paper, for the binary classification problem of whether it is a landslide or not, various validation methods were used from different perspectives. First, the most basic method is the *Accuracy*, as well as the *Precision* and *Recall* values for validating positive and negative samples, respectively [69]. Second, there are validation methods for binary classification problems such as F-value, *MCC*, and Kappa coefficient [49,70]. The third is the most common and visualized ROC curve, which evaluates model merit by measuring the area under the curve (AUC) [71,72]. Finally, SCAI was used to analyze the percentage of each class and the proportion of historical landslide points in each class [73]. The formulae for each of the above evaluation methods are as follows:

$$Accuracy = \frac{TP + TN}{TP + TN + FP + FN} \tag{14}$$

$$Precision = \frac{TP}{TP + FP} \tag{15}$$

$$Recall = \frac{TP}{TP + FN} \tag{16}$$

$$F1 = 2 \times \frac{Precision \times Recall}{Precision + Recall} \tag{17}$$

$$MCC = \frac{TP \times TN - FP \times FN}{\sqrt{(TP + FP)(TP + FN)(TN + FP)(TN + FN)}} \tag{18}$$

$$P_a = \frac{TP + TN}{TP + TN + FN + FP} \tag{19}$$

$$P_{\exp} = \frac{(TP + FN)(TP + FP) + (FP + TN)(FN + TN)}{\sqrt{(TP + TN + FN + FP)}} \tag{20}$$

$$\text{Kappa} = \frac{P_a - P_{\exp}}{1 - P_{\exp}} \tag{21}$$

where *TP*, *FP*, *TN*, and *FN* are true positive, false positive, true negative, and false negative, respectively.

The value domain of *Accuracy*, *Precision*, *Recall*, and F-value is from 0 to 1, and a greater value results in stronger performance. For *MCC* and Kappa coefficient, the results are between $-1$ and 1, and again, larger value means better.

The ROC is drawn according to the "*Sensitivity*" and the "1-*Specificity*" [74]:

$$Sensitivity = \frac{TP}{TP + FN} \tag{22}$$

$$Specificity = \frac{TN}{TN + FP} \tag{23}$$

The value domain of AUC ranges from 0.5 to 1, and the better performance is reflected by higher values [75].

The *SCAI* is calculated by the following Equation:

$$SCAI_i = \frac{P_A}{P_{HGP}} \tag{24}$$

where, *i* is the landslides susceptibility class, $P_A$ is percent for each class, and $P_{HGP}$ is the proportion of historical landslide points in that class to total points. This method can show the density of historical landslide points in each class. The higher classes should have lower *SCAI* values.

## 4. Results

### 4.1. Factors Multi-Collinearity Analysis and Importance

The collinearity analysis between these influencing factors at 95% confidence level is illustrated in Table 3. The results indicate that no multicollinearity relationship existed among factors (all the *VIF* values were below 7). That means the selected 17 factors can be adopted simultaneously for these neural network models.

**Table 3.** Multi-collinearity analysis results of explanatory variables.

| Explanatory Variable | Multi-Collinearity Statistics | |
| --- | --- | --- |
| | *VIF* | *Tolerance* |
| Lithology | 1.533 | 0.652 |
| Geological age | 1.234 | 0.810 |
| Faults | 1.666 | 0.600 |
| Elevation | 1.775 | 0.563 |
| Slope | 2.911 | 0.343 |
| Aspect | 1.078 | 0.928 |
| Plan curvature | 1.502 | 0.666 |
| Profile curvature | 1.483 | 0.674 |
| Land cover | 1.249 | 0.801 |
| NDVI | 1.193 | 0.838 |
| Roads | 1.122 | 0.891 |
| Rivers | 1.150 | 0.869 |
| AAP | 1.864 | 0.537 |
| SPI | 6.085 | 0.164 |
| STI | 5.132 | 0.195 |
| TRI | 3.257 | 0.307 |
| TWI | 4.802 | 0.208 |

The neural networks are black-box models, the factors importance ranking cannot be performed by them. Therefore, we input the training set into the RF model for the analysis of factor importance study. Figure 9 shows the radar plot of factor importance, the importance value of elevation is much more than other factors that means elevation is highest priority for development of landslides. As the most important trigger factor for landslides in the study area, APP has an importance of more than 0.7, second only to elevation. Plan curvature and NDVI are the third tier (importance value > 0.6), while the distance to rivers and SPI have little effect on landslides (importance value < 0.4).

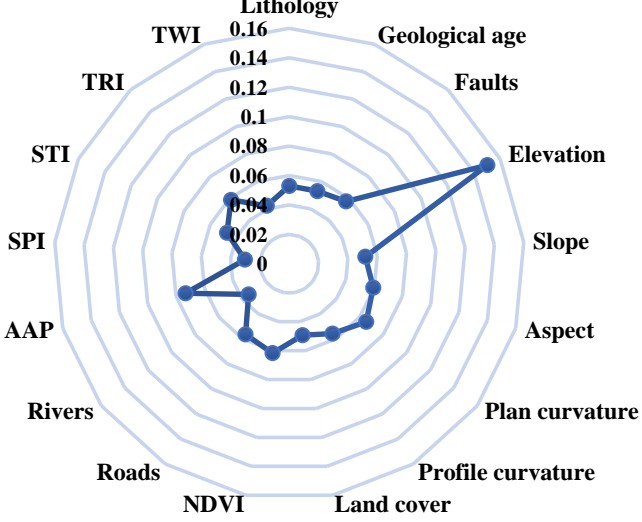

**Figure 9.** The radar plot of susceptibility factors importance of the landslides.

### 4.2. The Optimization Process of TPE

The principle of TPE optimization is to continuously loop through searching for the minimum objective function value, therefore, this paper iterates with accuracy as the return value of TPE optimization for 500 iterations, and its optimization process is shown in Figure 10. The minimum return value of DNN is −0.7535, which is generated after 367 iterations, while the initial return value of RNN is the largest and converges to −0.7604 after 258 iterations, and the return value of CNN is the smallest, reaching −0.7708 after 358 iterations. Overall, CNN has the best TPE optimization effect, and RNN and DNN have similar effects. If combined with the analysis of optimized initial values, RNN is slightly better than DNN.

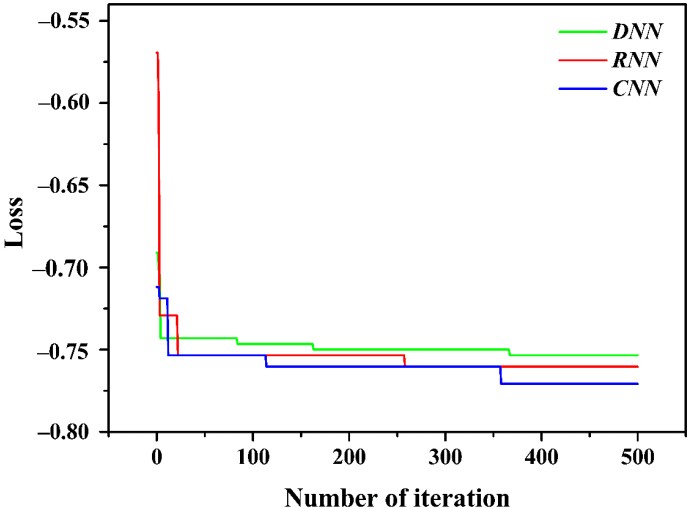

**Figure 10.** The TPE optimization loss in the iterative process of three typical neural network models.

Table 4 shows the hyperparameter results of TPE optimization for the three neural network models. The optimization results of DNN and RNN are very similar, including the neurons (units) in the hidden layer, dropout rate, batch size, and only the epoch values are different. Moreover, Adam algorithm is the best optimizer in DNN and RNN, which proves its universality to some extent. For the CNN, the kernel size is increased to 6 and the RMSProp algorithm is chosen for the optimizer, which ensures less oscillation in the return value during optimization and also makes the network function converge faster.

**Table 4.** The TPE optimization result of hyperparameters in the DNN, CNN, and RNN models.

| Model | Hyperparameter | Initial Value | TPE Optimization Result |
|---|---|---|---|
| DNN | Units | 35 | 96 |
| | Dropout rate | 0 | 0.395 |
| | Batch size | 1 | 38 |
| | Epoch | 50 | 42 |
| | Optimizers | Adam | Adam |
| RNN | Units | 50 | 95 |
| | Dropout rate | 0 | 0.357 |
| | Batch size | 1 | 42 |
| | Epoch | 50 | 69 |
| | Optimizers | Adam | Adam |
| CNN | Fitters | 20 | 20 |
| | Kernel size | 3 | 6 |
| | Pooling size | 2 | 2 |
| | Units | 50 | 69 |
| | Dropout rate | 0 | 0.177 |
| | Epoch | 100 | 177 |
| | Optimizers | Adam | RmsProp |

### 4.3. Landslide Susceptibility Assessment and Mapping

In this study, three typical neural network models were constructed using the Keras library (Version 2.3.1) in Python 3.7 environment. Keras is a high-level API for TensorFlow 2 that facilitates machine learning especially deep learning through easy operation [76]. The different models constructed based on the training set predicted the probability of occurrence of landslides (i.e., susceptibility) for each pixel in the study area, respectively. Coupled with MATLAB 2018b and ArcGIS 10.6 softwires, we plotted the LSMs thus visualizing the spatiality for susceptibility. For better visualization of LSMs, they were reclassified via NBM to 5 levels (Figure 11).

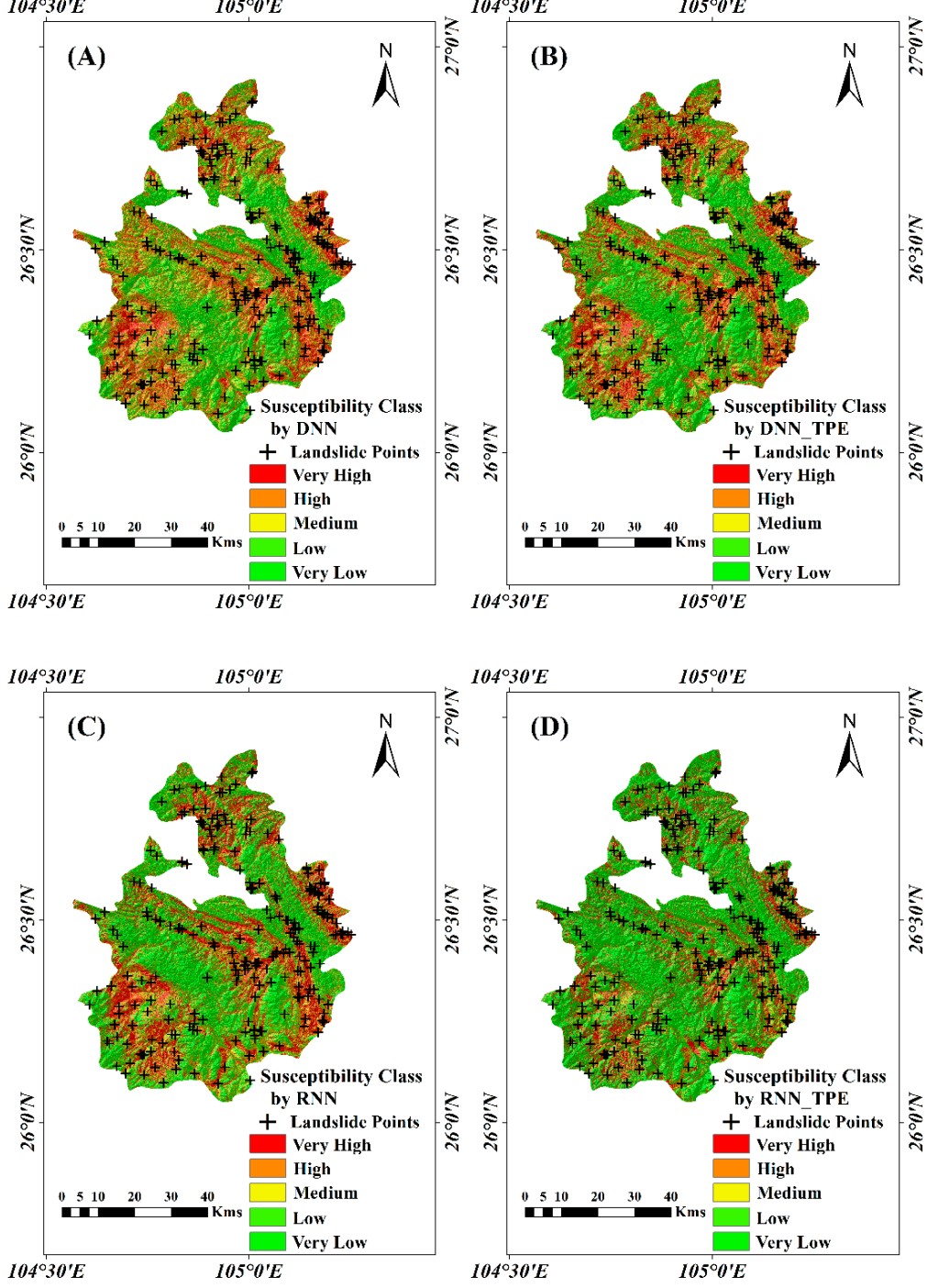

**Figure 11.** *Cont.*

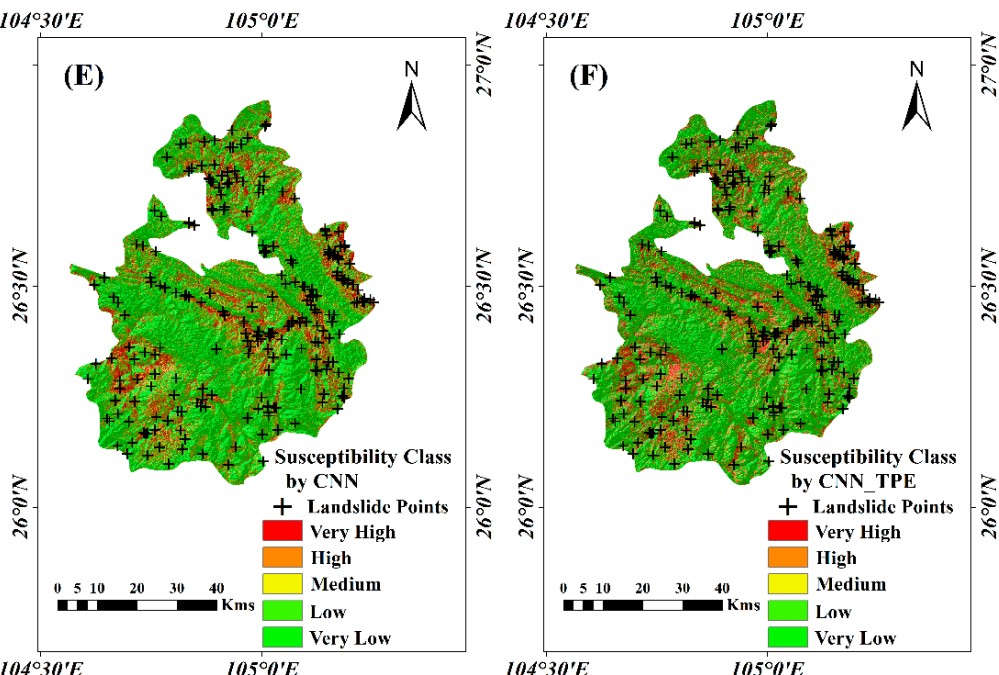

**Figure 11.** Landslide susceptibility maps based on (**A**) DNN, (**B**) DNN_TPE, (**C**) RNN, (**D**) RNN_TPE, (**E**) CNN, and (**F**) CNN_TPE.

The distributions of high susceptibility areas of different models are similar, all of them are near faults, and the lithology is mostly claystone, sandstone, and basalt, while for the topography, the greater the slope, the higher the probability of landslide occurrence. In terms of elevation, high susceptibility areas are mostly concentrated in mid-high elevation areas rather than in very high elevation areas. The main reason is that human activities in this elevation range may change the surrounding geological environment and thus affect the occurrence of landslides, which matched the actual situation with high reliability of the results. For lithology, limestone and dolomite are dense and hard blocky rocks, which are brittle and have great shear strength and can withstand large shear forces without deformation, while claystone, mudstone, and basalt have much clay or gravel soil, which have plasticity and relatively low shear strength and are easily deformed and landslides occur. For the slope, the greater the slope, the greater the potential energy of the landslides, the higher the sliding speed, and the landslides are more likely to occur induced by external forces. The results will provide an important reference for the zoning of landslide susceptibility and the setting of disaster prevention policies in Shuicheng County.

To facilitate comparison between models, we calculated $P_A$ and $P_{HGP}$ in each LSM, which is the most important step required to calculate SCAI values, as shown in Figure 12. For the DNN model, the area share of each class is very close, but the distribution of landslide points is obviously concentrated in the high-class area. The TPE optimization did not change the structure of the distribution of each class in the DNN model, but the number of landslide points in the "very high" class increases, which can reflect the optimization effect of TPE. For RNN, the grade distribution is U-shaped, with "very low" and "very high" accounting for 32.79% and 21.83%, respectively, and after TPE optimization, the percentage of "very low" reached 52.43%. Meanwhile, the "very high" class of RNN has the highest $P_{HGP}$ (47.08%), which decreases after TPE optimization, but the "high" class has a significantly higher $P_{HGP}$. The rank distribution of $P_A$ in CNN models decreases from "very low" to "very high", with "very high" accounting for nearly half (46.55%), and the CNN has the worst performance for $P_{HGP}$ results. By TPE optimization, the percentage of extreme classes is significantly increased, while the $P_{HGP}$ of high-class regions was increased and low and medium classes were decreased, which indicates that the TPE algorithm has good effect on the optimization of the CNN model.

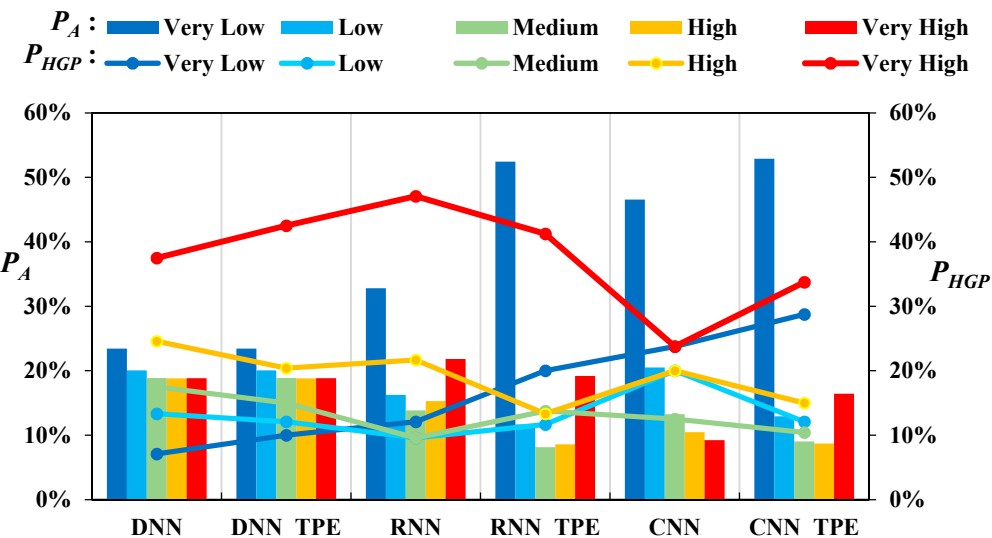

**Figure 12.** The statistics of each class of LSMs constructed by the models (PA is the proportion of each susceptibility class; PHGP is the proportion of historical landslide points in each class in each LSM).

### 4.4. Model Validation and Comparison

Properties and applicability of these models need to be validated and evaluated by different perspectives and methods. Table 5 displays validation results for different models. The Accuracy values of the models in descending order is DNN_TPE, CNN_TPE, CNN, RNN, DNN, RNN_TPE, which indicates that the TPE algorithm significantly optimizes the accuracies of DNN and CNN models, but has no effect on RNN. Differentiated into Precision and Recall analysis, TPE optimization significantly improves the performance of three typical neural network models for the correct rate of positive sample prediction according to the metric "Precision". For the ability to capture positive samples represented by Recall values, RNN has an obvious advantage over other models, but TPE optimization has a significant negative effect on RNN, while the other two models are improved. Moreover, F1 focuses on the balance of Precision and Recall, it can be obtained that RNN and TPE optimized DNN models are better for positive samples, while CNN model is the worst. MCC and Kappa coefficient are all indicators of the comprehensive performance of the model. Additionally, the MCC and Kappa coefficients are closer to the judgment of Accuracy.

**Table 5.** Model validation results using multiple methods.

|           | **DNN** | **DNN_TPE** | **RNN** | **RNN_TPE** | **CNN** | **CNN_TPE** |
|-----------|---------|-------------|---------|-------------|---------|-------------|
| Accuracy  | 0.715   | 0.743       | 0.719   | 0.712       | 0.722   | 0.733       |
| Precision | 0.701   | 0.733       | 0.676   | 0.697       | 0.735   | 0.745       |
| Recall    | 0.750   | 0.764       | 0.840   | 0.750       | 0.694   | 0.708       |
| F1        | 0.725   | 0.748       | 0.749   | 0.722       | 0.714   | 0.726       |
| MCC       | 0.432   | 0.487       | 0.451   | 0.425       | 0.445   | 0.466       |
| Kappa     | 0.431   | 0.486       | 0.438   | 0.424       | 0.444   | 0.465       |

Figure 13 illustrates six ROC curves for these models. The AUC value of RNN is much higher than the similar values of DNN and CNN. Whereas after TPE optimization, the AUC of RNN_TPE in turn has a slight decrease, the performance of both DNN_TPE and CNN_TPE improves, especially DNN, by 4.6%. This also shows the optimization effect of TPE in the DNN and CNN models, especially the DNN model, but does not have the desired effect on the RNN model.

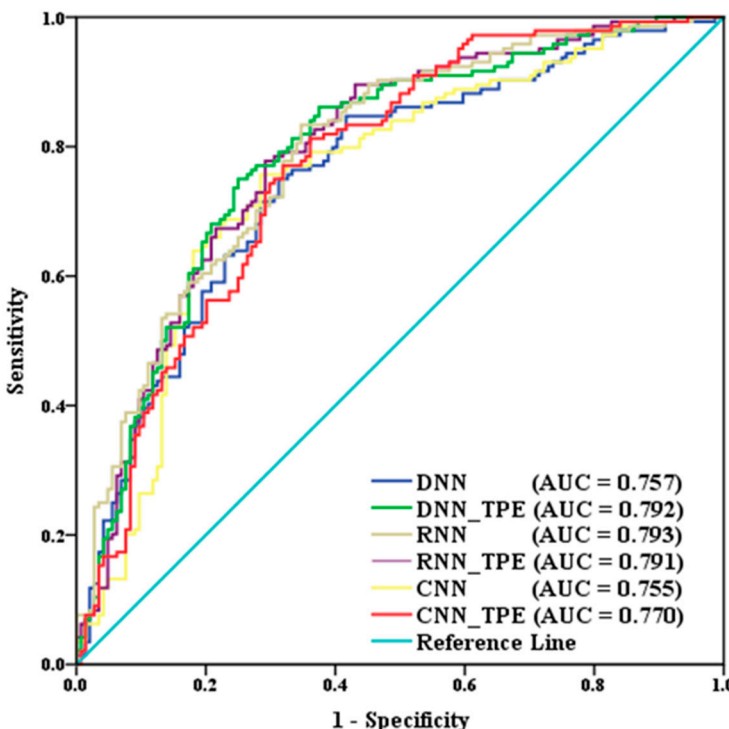

**Figure 13.** The ROC curves of the neural network models.

Table 6 presents the SCAI value of three typical neural network models before and after TPE optimization. Overall, different models show a trend of lower values for higher classes, with DNN and RNN models performing better for lower classes, while for higher classes, each model has good performance, especially the CNN model. TPE optimization's advantage is not obvious in the determination of SCAI values results, which may be caused by the difference in the ranking of different prediction results.

**Table 6.** The SCAI results of three typical neural network models before and after TPE optimization.

| SCAI ＼ Model Class | DNN | DNN_TPE | RNN | RNN_TPE | CNN | CNN_TPE |
|---|---|---|---|---|---|---|
| Very low | 3.3079 | 2.3431 | 2.7134 | 2.6216 | 1.9600 | 1.8397 |
| Low | 1.5043 | 1.6599 | 1.6960 | 0.9992 | 1.0252 | 1.0706 |
| Medium | 1.0784 | 1.2582 | 1.4446 | 0.5926 | 1.0597 | 0.8669 |
| High | 0.7650 | 0.9212 | 0.7055 | 0.6436 | 0.5241 | 0.5801 |
| Very high | 0.5022 | 0.4431 | 0.4636 | 0.4650 | 0.3882 | 0.4871 |

## 5. Discussion

Up to now, many scholars have conducted studies on LSA by using different methods and comparing the strengths and weaknesses between models [77–79]. This paper aims to provide an introduction for application and comparison of three typical neural network models (DNN, RNN, CNN) in LSA, and to optimize their hyperparameters using TPE algorithm in order to get better prediction accuracy and performance. Before training of these models, for the preparation of the sample set, due to the insufficient number of positive samples, this study proposed to use the hybrid ensemble oversampling and undersampling techniques, doubling the positive samples and matching an equal number of negative samples to meet the need for sample balancing. Then, the multiple collinearity analysis was performed on the influencing factors, which proved that the 17 factors were independent of each other and could be input into the mod-

els simultaneously. In addition, the RF model was introduced to compare the factors importance, and the LSM generated by combining multiple models, it is obvious that the high-susceptibility regions mostly distributed in bands along fault zones, and the influence of elevation on landslides is much higher than other factors. As the main predisposing factor leading to landslides in the study area, the importance of AAP is second only to elevation, which means that the hydrological conditions of geotechnical bodies cannot be ignored in the occurrence of landslides. Combined with LSMs analysis, the high susceptibility area is mainly concentrated near the faults, which is the structure where the strata or rock body is significantly displaced along the rupture surface, and the slope near the fault is also high, and the results are consistent with the actual landslide patterns. More high susceptibility areas are in claystone, sandstone, and basalt, which is also consistent with our statistics on the actual lithologies that are more prone to geological hazards. The local authorities should also propose appropriate policies based on the results of LSMs to focus on the protection of high susceptibility areas and to restrict their development works and activities. The discussion on the importance of factors can also provide help and reference for the forecasting and early warning of landslides for related departments.

Due to the black-box property of neural network models, it is impossible to intervene in the model operations, tuning hyperparameters in the preparation phase of the model is an important tool to improve the model performance. In this paper, the TPE algorithm was used to optimize the hyperparameters, combining the optimization results (Table 4) with the validation results of the models (Table 5 and Figure 13). For DNN, it directly passes the input layer data to the hidden layers and finally outputs the results, while in CNN, after convolution and pooling, the data is then passed through the fully connected layer, and although the accuracy is similar to DNN, the results show polarization, i.e., the model does have higher confidence in the prediction result. As for the RNN model, its complex recurrent structure of the hidden layer can make full use of the data information, and the performance is also the best. For TPE optimization, the number of samples input to the model in each iteration is controlled by increasing the batch size, but the epochs do not need to increase with it, which may still be influenced by the small sample size. The increase of neurons improves the model accuracy, but also sets a higher dropout rate for reducing overfitting, and for the choice of optimizer, the robustness of the Adam algorithm can be proved. In fact, the TPE optimization result of RNN is very similar to DNN, but the optimization effect of RNN is not improved as we expect, and the RNN is also the earliest convergence in the optimization process, which may also be a performance that cannot be effectively optimized (Figure 10), probably because the TPE optimization is not applicable to the complex, recurrent hidden layer structure in the RNN model. For CNN, its unique hyperparameters in which the kernel size increases from 3 to 6, as the convolutional kernel size increases, the receptive field increases and better features are obtained, which does not cause a large computational effort to the extent that it takes too long due to the small amount of computation. For the fully connected layer, the increase in neurons is small, and the overall number of epochs increases while the risk of overfitting is reduced by increasing the dropout rate, and the optimizer is replaced with RMSProp, this allows for less oscillation of the return value during optimization. The assessment effect of CNN model is also greatly enhanced by tuning the parameters.

The LSM produced by each model was combined and the statistical results of each class compared (Figures 11 and 12). Overall, the spatial distribution of classes for the three typical neural network models is similar, but the proportion of each class is significantly different. The DNN model is basically all divided into quintiles, and the $P_A$ values are also decreasing from higher to lower classes; the class distribution of RNN shows a slight polarization, while the distribution of $P_A$ is better; for CNN, the very low susceptibility class accounts for nearly 50%. The accuracy of DNN and CNN is similar by multiple methods verification, while RNN is the best (AUC = 0.793), which indicates that the recurrent structure of RNN can fully utilize the sample information

for operation. After TPE optimization, the class structure of DNN and CNN did not change much, but the distribution of PAs was more reasonable. In addition, The TPE optimization significantly improves the accuracy of the DNN and CNN (3.92% and 1.52%, respectively), and the AUC values of these two models improved by 4.62% and 1.99%, respectively, the performance was significantly improved for both models, especially for the DNN model, which demonstrated the optimization effect of TPE in the DNN and CNN models. For RNN, the TPE optimization makes the polarization of the susceptibility values more significant, but the distribution of $P_A$ becomes worse, so that the overall capability is not improved.

Compared with research of LSMs constructed by other methods, the accuracy of the DNN_TPE or RNN models is higher than frequency ratio (AUC = 0.75), weight of evidence (AUC = 0.76) [16], and LR (accuracy = 0.742, AUC = 0.79) [31], similar to the CNN model proposed by [41] (accuracy = 0.742, AUC = 0.80), but lower than the RNN model (accuracy = 0.762, AUC = 0.843) [40]. The reason for the difference in model accuracy with similar architecture is that the data input to each model is different, including sample size, selection of factors, etc., and cannot be directly compared. In comparison with our previous related study in Shuicheng County [36,46], the AUC value of the DNN_TPE or RNN models is higher than Bayesian network (AUC = 0.785), close to gradient boosting decision tree (AUC = 0.796), but lower than the RF model (AUC = 0.845). This result is similar to the findings of [27], although the neural network models are more advanced, the tree structured model performs better for one-dimensional data processing and classification. In addition, it may also be due to the neural network models requiring too many parameters to tune, which limits the accuracy of the model. Integrating the above discussion, we trained and validated all three models before and after the optimization in order to reflect the effect of TPE optimization and better reflect the significance of TPE optimization. The assessment framework proposed in this paper satisfies the need for accuracy, can provide guidance for disaster prevention and control, and also provides new methods and optimization strategies for LSA research, which has certain practical and theoretical significance.

Figure 14 plots variation curves of the accuracy and loss function values for the training and validation sets during the epochs of the six model runs. Since the purpose of model fitting is to continuously search for the minimum value of the loss function of training set, the training set loss and accuracy of each model are monotonically decreasing and increasing respectively with the iterative process, so we need more to observe the changes in the validation set. The accuracy of validation set of DNN_TPE has a higher decreasing slope in the initial stage than DNN, indicating that TPE optimization has a very intuitive effect on the simple fully connected layer structure. For RNN, the TPE optimization has little effect on the accuracy of the validation set, but the loss value becomes more volatile and rises with the epoch increases, which has generated the risk of overfitting, and the TPE optimization does not produce the expected effect. In fact, CNN_TPE also has the risk of overfitting, which may be caused by the small number of samples, but it can be clearly seen that the accuracy of the validation set significantly improved, and the LSA generated by CNN_TPE meets the requirement of accuracy from the perspective of the actual results and the distribution of historical landslide points. Reducing overfitting is still a problem that needs to be solved in future research and means such as reducing the epoch or increasing the dropout rate can be considered.

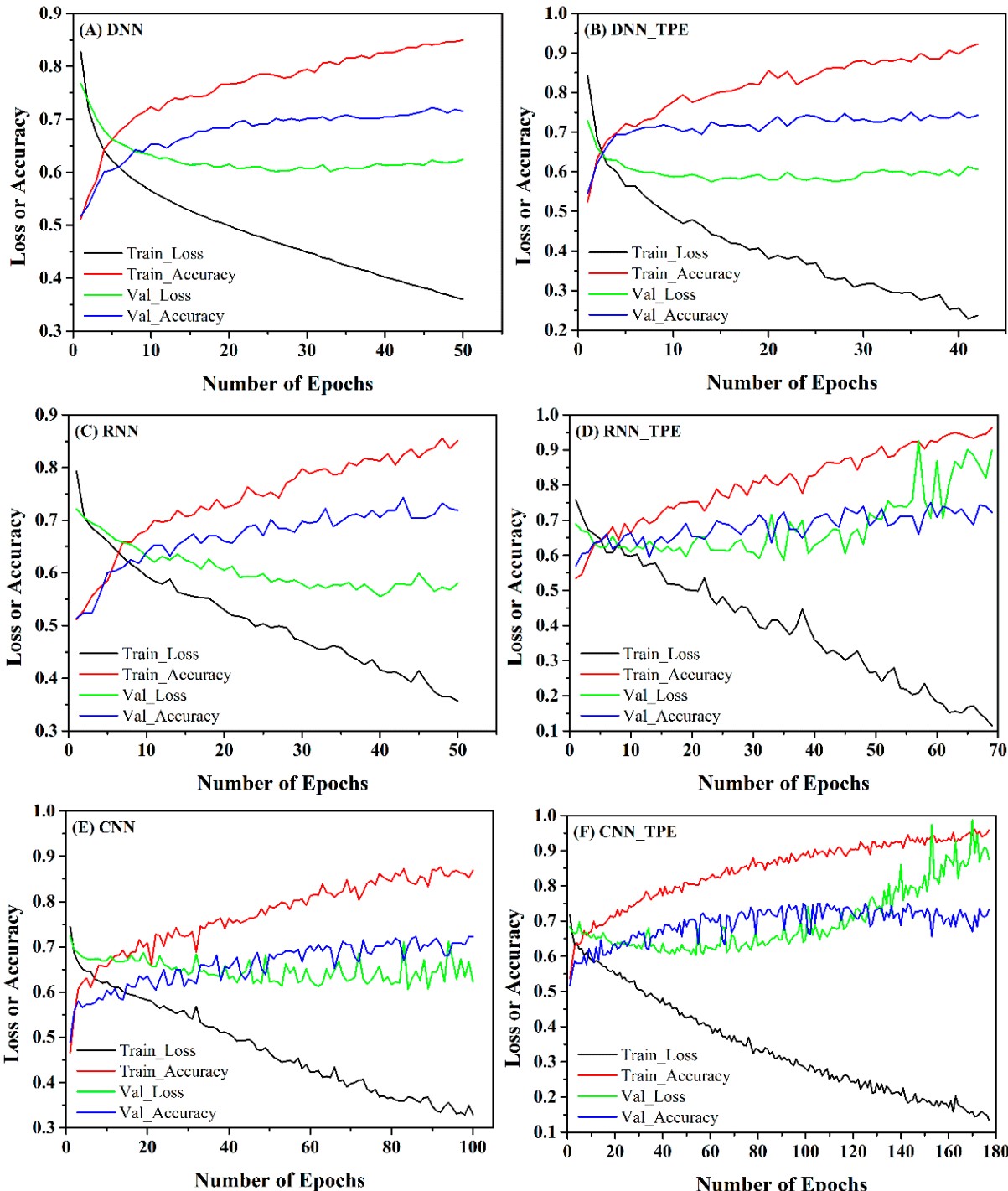

**Figure 14.** The loss and accuracy of the training and validation sets during the epochs of each model: (**A**) DNN, (**B**) DNN_TPE, (**C**) RNN, (**D**) RNN_TPE, (**E**) CNN, and (**F**) CNN_TPE.

## 6. Conclusions

Landslides pose a constant threat to the lives and property of mountain people and may also cause geomorphological destruction such as soil and water loss, vegetation destruction, and land cover change. The work on the assessment of landslide susceptibility is particularly important. The main purpose of this paper was to introduce TPE algorithm for hyperparameter optimization of three typical neural network models for landslide susceptibility assessment in Shuicheng County, China, as an example, and to compare the differences of predictive ability among the models in order to achieve higher applica-

tion performance, and the susceptibility assessment was carried out by extracting LSM. First, 17 influencing factors of landslide multiple data sources were selected for spatial prediction. For the problem of imbalanced sample and small sample size, hybrid ensemble oversampling and undersampling approaches were used to double the sample size and randomly split into training and validation sets. Multi-collinearity analysis was carried out for influencing factors, and RF model was used to perform factor importance ranking. Second, DNN, RNN, and CNN models were adopted to predict the regional landslides susceptibility, and the TPE algorithm was used to optimize the hyperparameters, respectively, to improve the assessment capacity. Finally, to compare and validate the predictive performance of the models, several objective measures of the Accuracy, Precision, Recall, F-value, MCC, Kappa value, ROC curve, and SCAI were used for evaluation. The results show that the high-susceptibility regions mostly distributed in bands along fault zones, where the lithology is mostly claystone, sandstone, and basalt. The selected 17 factors have no co-linearity problems, and elevation has the strongest influence on landslides, followed by precipitation. The DNN, RNN and CNN models all perform well in LSM, especially the RNN model, which has an AUC value of 0.793. The TPE optimization significantly improves the accuracy of the CNN and DNN but does not improve the performance of the RNN. In summary, our proposed RNN model and TPE-optimized DNN and CNN model have robust predictive capability for landslide susceptibility in the study area and can also be applied to other areas containing similar geological conditions. In future research, the application of TPE optimization to different neural network models and their related variants can be further improved, and the evaluation performance among different machine learning models can be compared and analyzed to a greater extent.

**Author Contributions:** Conceptualization, G.R.; data curation, G.R. and Y.Z.; formal analysis, G.R. and K.L.; funding acquisition, J.Z. and T.L.; methodology, G.R. and Y.S.; writing—original draft, G.R.; writing—review and editing, Z.T. and X.L. All authors have read and agreed to the published version of the manuscript.

**Funding:** This research was funded by the National Key Research and Development Program of China (2018YFC1508804), the Key Scientific and Technology Program of Jilin Province (20170204035SF), the Key Scientific and Technology Research and Development Program of Jilin Province (20180201033SF), the Key Scientific and Technology Research and Development Program of Jilin Province (20180201035SF), and the National Natural Science Fund for Young Scholars of China (41907238).

**Institutional Review Board Statement:** Not applicable.

**Informed Consent Statement:** Not applicable.

**Data Availability Statement:** The codes and data for this article can be freely available at https://github.com/ronggz728/DNN_RNN_CNN_TPE (accessed on 27 October 2021).

**Conflicts of Interest:** The authors declare no conflict of interest.

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
