# Peer review of "Comparison of Tree-Structured Parzen Estimator Optimization in Three Typical Neural Network Models for Landslide Susceptibility Assessment"

_remotesensing, doi:10.3390/rs13224694_

Round 1

Reviewer 1 Report

The reviewed manuscript is a comparison of susceptibility assessment using three different Neural Networks. In this work, the authors have consolidated all the slides into geo-hazards, then applied different NNs with different levels of optimization. The whole process is a replication methodology that is well known and has been redundant in the literature. It utilizes different factors, selects variance inflation factor to train and compares accuracy measures as result.

  1. Although landslides data collection is somewhat part of Remote Sensing, the work heavily falls under Geomatics, GIS, Data Science and Earth Engineering works and does not fit the journal’s scope. As GIS and NN are both mature technologies, this paper shows poor replication with very low implications compared to literature for such high efforts.
  2. Although there is a massive effort done by authors, the fundamental on geohazard is not well expressed. Also, NN has been performed which obviously will give certain results after feeding some data, but authors need to understand this process and how it is related to reality and literature of geohazard. Here the results show elevation plays important role in geohazard than after rainfall, but usually its combination of slope and soil type that matters most. A thorough investigation and discussion are a must for localized data-based results.
  3. First, there are countless papers published over the last few decades since GIS became available looking at similar conditioning factors such as NDVI, Slope, aspect, profile, rainfall, etc. to map landslide susceptibility. They all read the same as this paper. They use methods A to Z to produce maps of similar accuracy and conclude one better than others. Certainly, the maps have value for the very local scale for which they have been created. Changing factors, data and method will have a very small impact on the accuracy, and implications are the same. If you see the following authors, all the contributions are the same with changing areas or methods:
    • https://www.semanticscholar.org/author/B.-Pradhan/143620129
    • https://www.semanticscholar.org/author/S.-Lee/39847097
    • https://www.semanticscholar.org/author/H.-R.-Pourghasemi/2858616
    • https://www.semanticscholar.org/author/D.-Bui/2382463
    • https://www.semanticscholar.org/author/H.-Hong/27099181
    • https://www.semanticscholar.org/author/B.-Pham/144109958
    • https://www.semanticscholar.org/author/H.-Shahabi/48673769
    • https://www.semanticscholar.org/author/B.-Ahmad/32082590
    • https://www.semanticscholar.org/author/A.-Shirzadi/23155839
  4. Papers from the above authors have been in the past decade. Certainly, the maps have value for the very local scale for which they have been created. However, the results are very specific to the given datasets and have only the conclusion of one better than another which does not give new information for the broader scientific community, and this does not warrant a publication.
  5. The authors’ use of NN is a well-known state of art application and here few variabilities on the NN has been presented. There are many such variations, each can be shuffled to the same methodology to compare accuracies.
  6. Here the AUC are below 0.8 which is fine but not good compared to others in the literature. And here factors can change and even improve if you use a simple decision tree, NN, SVM or simple probability like the weight of Evidence or frequency ratio. methods depending on what is fed. Thus, such a replication work should never be presented as an academic work unless it's a breakthrough application and has huge implications. Other such works have not been touched in an introduction or discussed their accuracies to compare findings creating a bit biased paper.
  7. The abstract, body, and works are precisely the same for countless papers where one can simply replace methods and values to get slightly different accuracy. This will not change any decision-making for disaster management in the area. This pattern has caused falsehoods to the new researchers in geology and mapping and has been misleading them to do real research and novel publication works. See the influence of those above authors.
  8. It is evident that graduate students need to have successful publications, but it should be some achievement to be proud of. This work is a good case study and has some implications in the area but cannot be an academic advancement in the field of geology. Again, these are technical skills to apply on a project site and should not be pursued as an academic thesis/publication by supervisors. Understanding soil, geology, and triggering thresholds with field and lab experiments are what research questions should answer that would be useful to those regions.
  9. The literature build in introduction needs substantial improvement. The author’s objective is to optimize their hyperparameters using the TPE algorithm to get better prediction accuracy and performance, which is not enough for academic publication in the journal.
  10. The abstract needs improvement to the point. See https://mitcommlab.mit.edu/cee/commkit/abstract/
  11. The introduction is not built well and there is no proper literature gap to focus on. All statements are well known and end with a methodology for Susceptibility Assessment with XYZ methods.
  12. Although all slides are a result of gravity, there are quite differences in them, debris and landslide mostly occur by the influence of rain while rockfall may not. So, combining all those might be somewhat an issue. See updated Varnes classification system  https://www.researchgate.net/publication/263340148
  13. Also, the debris slides seem quite large, taking centroid or starting point would be an idea? Need proper literature for these assumptions
  14. No proper research gap is established for this specific work. See https://mitcommlab.mit.edu/cee/commkit/journal-article/
  15. The area of study is quite small and sampling slides are quite low. Although many ML papers use these, for NN to work and assume better weights massive data are required. For the area, hazard points look quite high in the figs.
  16. Selection of study area as a county is not a good choice and here it’s quite curved with a gap. According to Acharya, 2018, study areas that have concentrated distribution of the landslide events can cause the map to be very biased and misleading https://www.researchgate.net/publication/322023909. Thus, the selection of must is watershed or ridges that is where gravity acts for such slides.
  17. Temporal and spatial resolution in the data selection process needs to be properly explained. Add these in tale 1.
  18. Also varying classes of continuous fields in table 1 impact the result very much. So why these intervals?
  19. Is equation 6 an area-based frequency ratio?
  20. Discussion somewhat points the problem: Up to now, many scholars have conducted studies on GSA by using different methods and comparing the strengths and weaknesses between models” but authors repeat the same. It only explains what NN plots show. A little bit different technical note/communication like Li et al, 2021 can be done in case large studies are conducted and authors want to communicate the findings https://doi.org/10.3390/s21155191.
  21.  The discussion needs improvement. Focus discussion first landslide patterns should be discussed, then susceptibility in details, the reality in ground reality, natural factors, policy, development works, and activities comparing parallelly with the result. Please read properly what discussion should be in the journal’s template and https://mitcommlab.mit.edu/broad/commkit/journal-article-discussion/
  22. Reporting accuracy and number from a computer model is output, not an outcome. Research work must have an outcome that has some information or implication.
  23. Random and unrelated citations. Most citations are only done based on keywords and not their conclusion drawn or the basis that the research gap is built. Please understand the concept of the citation properly, use the one you read, understand, and conclude your statement written. To go through 75 publications is not an easy task.
  24. Recent prediction works have also created black-box research which is not easily verifiable and available to readers. The data and model should also be provided for transparency and replication via a proper Zenodo or GitHub or official websites of lab/province/dept (e.g., https://northernchange.brown.edu/data-and-code/). As an open-access and publicly funded work, this should be considered by authors so that it can be evaluated, and there will be an actual use and implication. For details https://authorservices.taylorandfrancis.com/data-sharing/share-your-data/data-availability-statements/
  25. A minor ethical issue, authors in Asia add all the funding sources in related/unrelated publications. It does not seem like such Chinese competitive fellowships and funding were awarded for such replication susceptibility work. Also, why a Jilin province would fund a study in Guizhou province. As a general suggestion for the future, to improve the reliability and standard of research in Asia, unsolicited authorship and fundings should proceed with high integrity.

Wish you all the best for future research work.

Reviewer 2 Report

Dear authors,

Here to you my comments:

  • lines 51-53: please use a reference for this statement.
  • lines 61-63: you can easily make shorter this sentence. Some information are redundant. I suggest something like 'Geo-hazards susceptibility assessment (GSA) is a key component of the geo-hazards risk evaluation.'
  • line 87: you can add a very recent study using Machine Learning for Risk assessment, Novellino, A.Cesarano, M.Cappelletti, P.Di Martire, D.Di Napoli, M.Ramondini, M.Sowter, A.Calcaterra, D.. 2021 Slow-moving landslide risk assessment combining Machine Learning and InSAR techniques. CATENA, 203, 105317. https://doi.org/10.1016/j.catena.2021.105317
  • Probably the biggest flaw in the Introduction. I would expect you to provide a line on why your work is important. Something like, 'our novel technology contribute to improve landslide susceptibility products'. I think is important considering the foreword of your Introduction.
  • lines 116-117, I think rainfall is the main triggering factor for landslides only, not for all geohazards...
  • lines 118 and 139, the landslides definition encompasses rockfalls and debris flows...
  • Section 2.3. I think you must add some references on the NDVI index. It is quite a well established factor used to map landslides... The same applies for other indices used later like TWI...
  • I think the figure on Geological Time of your units is not needed, it does not add anything to your calculations..
  • Not sure how a single NDVI, SPI, STI map can be useful. For example, I think NDVI is powerful if looked over time as NDVI changes differently in vegetated and non-vegetated areas... Please provide example from the literature to back up your idea to use a single map for these parameters and not taking into account the time variation.
  • I would suggest to have formula (19) after (20) and (21) as you need Pa and Pexp to derive Kappa

Round 2

Reviewer 1 Report

The replies and revision are both a good effort from the authors.

Given that SI has a landslide related focus in susceptibility modelling, and has already published a similar paper that only reports accuracy for some combination of methods, this paper also falls on the domain. The replies completely ignore the comments that in the past many papers have been doing such replication works and these types of studies have problems, so should be carefully considered, rather than build replies based on those replication studies which should be stopped. All replies were others have done in the past and we will keep doing and even says that we will keep doing other methods.

Swapping study areas, use of different controls, and mixing different methods to report accuracy does not add any value in the real world. Simple methods and state of art SVM, NN etc are more than enough to do actual work and prevent lives, it has never been reported ever that increasing 1% accuracy by LSM studies made any local government make decisions.

The decision is up to the editorial to evaluate the current work.
